# Specific N-cadherin–dependent pathways drive human breast cancer dormancy in bone marrow

Garima Sinha[1,2], Alejandra I Ferrer[1,2], Seda Ayer[2], Markos H El-Far[1,2], Sri Harika Pamarthi[2], Yahaira Naaldijk[2], Pradeep Barak[4,5], Oleta A Sandiford[2], Bernadette M Bibber[1,2], Ghassan Yehia[3], Steven J Greco[2], Jie-Gen Jiang[4,5], Margarette Bryan[2], Rakesh Kumar[6], Nicholas M Ponzio[4,5], Jean-Pierre Etchegaray[7], Pranela Rameshwar[1,2]

The challenge for treating breast cancer (BC) is partly due to long-term dormancy driven by cancer stem cells (CSCs) capable of evading immune response and resist chemotherapy. BC cells show preference for the BM, resulting in poor prognosis. CSCs use connexin 43 (Cx43) to form gap junctional intercellular communication with BM niche cells, fibroblasts, and mesenchymal stem cells (MSCs). However, Cx43 is an unlikely target to reverse BC dormancy because of its role as a hematopoietic regulator. We found N-cadherin (CDH2) and its associated pathways as potential drug targets. CDH2, highly expressed in CSCs, interacts intracellularly with Cx43, colocalizes with Cx43 in BC cells within BM biopsies of patients, and is required for Cx43-mediated gap junctional intercellular communication with BM niche cells. Notably, CDH2 and anti-apoptotic pathways maintained BC dormancy. We thereby propose these pathways as potential pharmacological targets to prevent dormancy and chemosensitize resistant CSCs.

## Introduction

Breast cancer (BC) remains a clinical problem despite the development of new drugs, improved screenings, and early intervention (Berry, 2013; Miller et al, 2016). One of the major challenges is the ability of BC cells (BCCs) to survive for decades in a dormant state. Such a state enables the BCCs to exist in cycling quiescence, which supports immune evasion and resistance towards therapy (Braun et al, 2005, 2009; Gelao et al, 2013; Mao et al, 2014; Massague & Obenauf, 2016; Otvos et al, 2016; Tjensvoll et al, 2019). These dormant BCCs can eventually serve as the source of metastatic cancer decades after remission (Talmadge, 2007; Tivari et al, 2018; De Angelis et al, 2019). Shared functional and molecular similarities between dormant BCCs and healthy stem cells have led to the

designation of cancer stem cells (CSCs) (Carcereri de Prati et al, 2017; De Angelis et al, 2019).

BCCs show preference for BM, resulting in poor prognosis (Braun et al, 2005; Tjensvoll et al, 2019). Once in the BM, BCCs evoke the resident BM niche cells (such as macrophages, fibroblasts, and mesenchymal stem cells [MSCs]) to facilitate their transition into a dormant phase (Rao et al, 2004; Patel et al, 2012, 2014; Bliss et al, 2016; Sandiford et al, 2021). Investigations to identify specific markers for CSCs revealed continuous phases of cellular plasticity, which include dedifferentiation of BCC progenitors into BCCs with stem cell–like properties (Naume et al, 2007; Plaks et al, 2015; Wang et al, 2017; Tjensvoll et al, 2019; Lynch-Sutherland et al, 2020). Thus, it could be argued that CSCs may be defined as a cellular state that is dependent on the tissue microenvironment. Indeed, MSCs, which constitute the cellular niche within the BM, can induce BCC quiescence through gap junctional intercellular communication (GJIC) and, through their exosomal secretome (Patel et al, 2014; Bliss et al, 2016; Sandiford et al, 2021).

The connexin (Cx) family of proteins constitutes gap junction. Six connexins assemble to form connexon (hemichannel), which pairs with hemichannel on neighboring cell to form a pore-like structure/gap junction. GJIC is among the mechanisms by which BCCs directly interact with BM niche cells to allow for the exchange of molecules to sustain dormancy (Lim et al, 2011). Expression of Cx26, Cx32, and Cx43 has been reported during different stages of BC development (Sinha et al, 2020). Specifically, CSCs exhibit high expression of Cx26 and Cx43 in comparison to BC progenitors; however, only Cx43 was involved in gap junction formation (Patel et al, 2012). Interestingly, Cx26, independent of forming gap junction, can maintain CSC stemness by complexing with Nanog and Focal adhesion kinase (Thiagarajan et al, 2018).

Despite a critical role for Cx43 in BC dormancy within BM, its ubiquitous expression and involvement in key cellular processes such as hematopoietic regulation make this an unlikely druggable target to reverse GJIC (Taniguchi Ishikawa et al, 2012; Martins-Marques

[1]Rutgers School of Graduate Studies at New Jersey Medical School, Newark, NJ, USA   [2]Department of Medicine, Hematology/Oncology, Rutgers New Jersey Medicine School, Newark, NJ, USA   [3]Genome Editing Shared Resource, Office of Research and Economic Development, Rutgers University, New Brunswick, NJ, USA   [4]Department of Pathology, Immunology and Laboratory Medicine, Rutgers New Jersey Medical School, Newark, NJ, USA   [5]ONI, Linacre House, Oxford, UK   [6]Department of Biotechnology, Rajiv Gandhi Centre for Biotechnology, Kerala, India   [7]Department of Biological Sciences, Rutgers University, Newark, NJ, USA

Correspondence: rameshwa@njms.rutgers.edu

et al, 2015). Therefore, it is imperative to identify new molecular mechanisms leading to GJIC between BM niche cells and CSCs that will not result in adverse effects upon targeting. Given that Cx43 expression was highest in CSCs and such cells are difficult to eradicate, this study investigated the associated mechanism in Cx43-mediated GJIC between CSCs and BM niche cells to maintain BC dormancy. The long-term goal is to discern a strategy to effectively sensitize the dormant BCCs to chemotherapy and/or to identify new treatments.

Studies in nonmalignant models reported on a role for CDH2 (neuronal cadherin/CD325/Cadherin-2) to translocate Cx43 to the cell membrane (Wei et al, 2005; Matsuda et al, 2006). CDH2 belongs to the cadherin family and is involved in cell adhesion, asymmetric cell division, and pre-/postsynapsis in the central nervous system (Garcia-Castro et al, 2000). The role of CDH2 in cell migration and invasion has been reported for several cancer cells, including BCCs (Kim et al, 2000). During epithelial mesenchymal transition (EMT), cancer cells can morphed in CSC-like phenotype with increased CDH2 (Yang et al, 2017). CDH2 has five extracellular domains, a transmembrane region, and a conserved cytoplasmic domain (Zuppinger et al, 2000). The discussed role of Cx43 and CDH2 in CSCs, which are associated with a dormant phenotype, led us to investigate how CDH2 contributes to Cx43-mediated gap junction for the purpose of long-term cancer cell survival in the BM.

BCCs can acquire dormancy at secondary sites, including the BM (Boire et al, 2019). The microenvironment within an organ is crucial for BC dormancy acquisition and maintenance. In this study, we examined how BM microenvironmental cells interact with CSCs by applying a previously reported BCC hierarchy (Patel et al, 2012). The two key nonhematopoietic cells incorporated in the experimental model are stromal fibroblasts and MSCs, both of which can establish GJIC with CSCs for sustained dormancy (Lim et al, 2011; Patel et al, 2014).

We report on a specific role for CDH2 in forming GJIC between CSCs and BM niche cells. Interestingly, CDH2 and Cx43 expressions were similar in the subset of BCCs with abundance CSCs. Using in vivo and in vitro studies, we showed CDH2-associated pathways as potential drug targets to reverse dormancy and thereby chemosensitize the otherwise resistant dormant CSCs. Importantly, CDH2 knockdown blocked GJIC without altering Cx43, which is required for GJIC. Colocalization of CDH2 and Cx43 in BM biopsies of BC patients was similar to the cell lines, underscoring the significance of the findings. Overall, these findings advance our understanding of BC dormancy and highlight potential therapeutic strategies.

# Results

## Increased CDH2 protein in CSCs

The major question posed in this study is to determine the mechanisms of BC dormancy, specifically pathways involved in GJIC between CSCs and BM niche cells (Patel et al, 2012). We selected CDH2 and its associated pathways because CDH2 is required for expression of transmembrane protein Cx43, which occurs partly by intracellular interaction (Wei et al, 2005; Matsuda et al, 2006).

Because CSCs showed preference in GJIC with BM niche cells (Patel et al, 2012; Walker et al, 2019), we compared the relative expression of CDH2 in different BCC subsets, including CSCs.

We first analyzed unsorted BCCs and two BM niche cells (MSCs and fibroblasts) for CDH2 and Cx43 (control) in Western blots (Patel et al, 2012) (Fig 1A). We characterized MSCs by morphology, phenotype, and multilineage differentiation (Fig S1A–C). There were strong bands for CDH2 and Cx43 in the BCCs with a weak band for CDH2 in stromal cells (Fig 1A). We verified the weak CDH2 bands in MSCs and stroma by immunoprecipitation with anti-CDH2 followed by blotting for Cx43 (Fig S1D).

We previously established an exhaustive model of BC dormancy within BM stromal compartment, close to the endosteum (Patel et al, 2012). In this study, BCCs with high *Oct4a* expression demonstrated properties of dormancy, including stem cell function. There was independent validation of the model (Bliss et al, 2016; Walker et al, 2019). We acknowledge that other groups have reported on models of dormancy that uses the integrin family of proteins, including dormancy in lung, as well as an orthotopic mouse model (Lu et al, 2011; Abravanel et al, 2015; Albrengues et al, 2018). Our model recapitulates human dormancy within the cellular BM compartment, before bone invasion and dormancy established by gap junction with resident BM nonhematopoietic cells. Thus, our model expands on understanding of BC dormancy.

Next, we compared different BCC subsets for CDH2 with MDA-MB-231-pOct4A-GFP, which was under the control of pOct4a regulatory region. The GFP was under the control of *Oct4a* regulatory region, thus intensity of GFP would be proportional to *Oct4a* expression, as reported (Patel et al, 2012). We sorted different cell subsets, based on GFP intensity (Patel et al, 2012) (Fig 1B). qPCR indicated similar ($P > 0.05$) CDH2 mRNA among BCC subsets (Fig S1E). However, flow cytometry and Western blot indicated a direct correlation between membrane and intracellular CDH2 with BCC maturity (Figs 1C and D and S1E, respectively). In summary, CSCs (GFP[hi]) showed the highest CDH2 protein as compared with the other BCC subsets. Thus, the results linked CDH2 to cellular stemness.

## Role of CDH2 in GJIC between CSCs and BM niche cells

CDH2 was increased in CSCs, which show preference for GJIC with BM niche cells (Figs 1C and D and S1E). We therefore investigated a role for CDH2 in GJIC using loss and gain of function studies (Patel et al, 2012). We selected CSCs from MDA-MB-231-pOct4a-GFP, knockdown for CDH2 (red fluorescence protein, RFP), Cx43 (RFP), or scramble shRNA using the depicted gating scheme (Fig 1E). Among the four CDH2-shRNA clones, Western blot for CDH2 with unsorted and CSCs indicated that Clone B was most efficient (Fig S1G and H). We co-cultured CSCs with 7-amino-4-chloromethylcoumarin (CMAC) (blue) labeled stromal fibroblasts or MSCs for 72 h and then examined the CSCs for CMAC transfer by fluorescence microscopy (Fig 1F). The images (Figs 1G and H and S1I and J, top row with enlarged regions below) showing white areas indicate dye transfer (yellow CSCs + blue CMAC = white) with scramble shRNA and vehicle. This transfer was blunted with GJIC inhibitor, 1-octanol, which served as a positive control. The blunting effect of 1-octanol mirrored the results with Cx43 or CDH2 knockdown CSCs. Flow cytometric analyses for dye transfer with deep red-labeled stromal fibroblasts

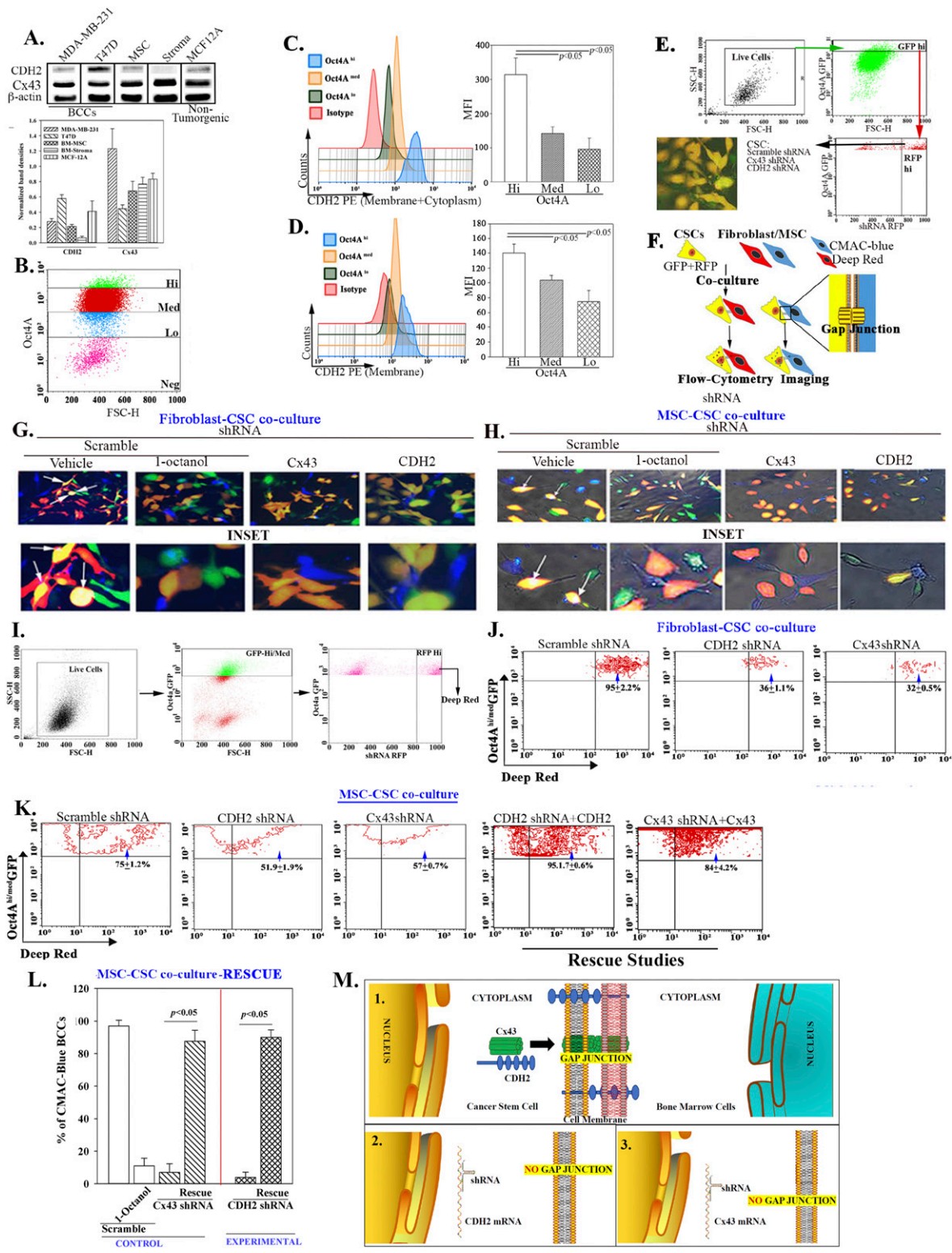

**Figure 1. CDH2 in gap junctional intercellular communication (GJIC).**
**(A)** Top panel: Western blot for CDH2 and Cx43 with whole-cell extracts from BCCs, stromal fibroblast, and MCF12A. The normalized band densities are shown below. **(B)** Shown is the gating scheme to select different BCC subsets with MDA-MB-231 cells with stable pEGFP1-Oct4A. **(C, D)** Representative histogram for flow cytometry for total (intracellular + membrane) (C) and membrane (D) CDH2 (left). The mean fluorescence intensities are presented ±SD for five independent analyses (right). **(E)** The gating scheme used to sort cancer stem cells (CSCs) (GFP), knockdown for CDH2 (RFP) or Cx43 (RFP). Lower left shows a representative image of the sorted cells (GFP + RFP = Yellow). **(F)** Diagram for the assay used to detect dye transfer from stromal fibroblast to CSCs. **(G, H)** MDA-MB-231-pOct4A-GFP were transfected with scramble-RFP

and MSCs showed the most immature BCC subsets (gating scheme for GFP[hi+med] BCCs shown in Fig 1I) with scrambled shRNA receiving 95% and 75% dye from fibroblasts and MSCs, respectively (Fig 1J and K). Such transfer was significantly ($P < 0.05$) reduced with CSCs, knockdown for CDH2 or Cx43 (Fig 1J and K).

Gain-of-function studies replaced CDH2 and Cx43 in the knockdown BCCs using an inducible expression system (Fig S1K and L). This induced CDH2 allowed for stable expression despite the shRNA. The rescued CSCs, when co-cultured with CMAC-blue–labeled MSCs, restored CSC's ability to establish GJIC with >80% dye transfer (Fig 1K, two right panels).

Dye transfer occurred between Oct4a[hi/med] and BM niche cells, and also between Oct4a[hi]/CSCs and MSCs (Fig 1J–L). Rescue of CDH2 and Cx43 in the knockdown BCCs resulted in dye transfer similar to scramble shRNA. Control with 1-octanol showed minimal dye transfer. Overall, CDH2 was necessary and sufficient to form GJIC connecting CSCs with stromal fibroblasts and MSCs (Fig 1M).

## Colocalization of CDH2 and Cx43 in BCC subsets

Other experimental system reported on intracellular colocalization between Cx43 and CDH2 as a requirement for Cx43 to be expressed on the cell membrane (Wei et al, 2005; Matsuda et al, 2006). Thus, we asked if CDH2 is in close proximity to CX43. Image stream of single unsorted BCCs, labeled for intracellular CDH2 (PE) and Cx43 (AF488), showed colocalization with a bell-shape distribution (Fig 2A). This suggested that colocalized CDH2 and Cx43 within the unsorted BCCs were heterogenous. To narrow the BCC subset with prominence colocalization, we selected CSCs with higher expression of CDH2 and repeated the image stream analyses (Fig 1B–D). The results indicated bright yellow fluorescence in the CSCs/Oct4a-GFP[hi] as compared with the other BCC subsets, indicating that colocalization was most efficient in CSCs (Fig 2B).

We validate CDH2-Cx43 colocalization using nanoscale imaging of individual BCCs (Fig 2C). MDA-MB-231 cells were colabeled with anti-Cx43-Alexa 647 (green) and anti–CDH2-PE (red) and then examined with a Nanoimager. Isotype control showed no background fluorescence whereas cells labeled with anti-Cx43 and CDH2 indicated intracellular and membrane (yellow) colocalization.

Next, we sought the regions of intracellular colocalization between Cx43 and CDH2 by colabeling for anti–CDH2-PE (red), anti–Cx43-Alexa 488 (green), and the one of following organelle-specific antibodies: anti–GRP78-Alexa405 (blue) for ER or anti–GM130-Alexa 405 (blue) for Golgi (G) (Fig 2D). Identifying these organelles will be important to fully decipher how Cx43 and CDH2 interact to establish dormancy and therefore lead to avenues in developing treatment. The white regions indicated colocalized CDH2 and Cx43 in both organelles: blue for organelle, red for CDH2, and green for Cx43. Next, we asked if intracellular colocalized CDH2 and Cx43 were relevant for Cx43 to reach the cell membrane. To address this, we preincubated BCCs with 5 µg/ml brefeldin A for 24 h, followed by labeling for Cx43 and CDH2 (Fig 2E). Brefedin A, which inhibits protein transfer from ER to Golgi, resulted in retention of both proteins in the cytosol (inset) and undetectable Cx43 on the cell membrane. In summary, these results demonstrated CDH2-Cx43 colocalization in the ER, G, and the cell membrane (Fig 2F).

## CDH2-Cx43 colocalization in BM biopsy of BC patients

We asked if BM biopsies of BC patients also show colocalized Cx43 and CDH2. We analyzed BM sections from 22 available BC patients by immunohistochemistry. The pathologist provided us with the linked demographics in the repository. Because some of the patients were treated, analyses of these samples would be important to determine if the residual BCCs have colocalized CDH2 and Cx43.

We performed triple labeling with anti–Cx43-AF-488 (green), anti–CDH2-PE (red), and anti–pan-cytokeratin-AF-405 (blue). We randomly selected two images from BC patients and hematological malignancy for Fig 3A with the remainder in Figs S2 and S3. The isotype control is shown in Fig S3C. Shown are colocalized CDH2 and Cx43 in cytokeratin+ cells (green + red + blue = white areas) as well as in some sections from hematological malignancies (Fig S3B). However, the percentages of colocalized CDH2 and Cx43 in hematological malignancies were significantly ($P < 0.05$) less than the number of slides with positive colocalized CDH2/Cx43 from BC patients (Fig 3B). One slide from hematological malignancy (#34) expressed pan-cytokeratin, suggesting this subject might have an undiagnosed solid tumor (Fig S3B). Analyses of slides from benign tumors showed weak evidence of Cx43-CDH2 colocalization (Fig S3A). The data indicate predominance colocalization of CDH2 and Cx43 in the BM of BC patients but not in benign BC sections.

## Interaction between CDH2 and Cx43 in BCCs

We asked of colocalized CDH2 and Cx43 interact. In silico analyses with the Fast Fourier Transform based protein docking program, ZDOCK, using the X-ray crystallographic structure of CDH2 ectodomain (grey) and Cx43 cytoplasmic domain indicated five interaction

---

(control), CDH2-shRNA-RFP, or Cx43-shRNA-RFP and then co-cultured) in four-well chamber slides at 1:1 ratio with BM stromal fibroblasts or mesenchymal stem cells (MSCs), respectively. The two latter cells were labeled with CMAC (blue). After 72 h, the cells were imaged for dye transfer using EVOS FL2 Auto 2 (200×). Parallel co-cultures with scramble shRNA contained 300 µM 1-octanol. Dye transfer is shown in the white areas (arrows). Larger images of the top images are shown below. **(G, H, I)** Gating scheme: the imaging studies for dye transfer in "(G, H)" were repeated by flow cytometry as follows: BM fibroblasts and MSCs were labeled with deep red dye. The live cells were gated based on FSC and SSC. After this, we selected RFP[hi] (shRNA) BCCs within the Oct4a-GFP[hi/med] subset then analyze for deep red dye transfer into the selected BCCs. **(J)** Flow cytometry representing the % dye transfer in fibroblast (included in the respective quadrant). **(K)** Flow cytometry representing the % dye transfer in MSCs (included in the respective quadrant). The last two panels show that CDH2 and Cx43 were rescued in the respective knockdown CSCs and then used for dye transfer from MSC into Oct4aGFP[hi/med] BCCs by flow cytometry. The % dye transfer is shown within the respective quadrant. **(K, L)** The rescue studies in (K) were repeated by imaging except for labeling MSCs with CMAC-blue dye. The cells showing dye transfer were imaged with the EVOS FL Auto 2, 200× magnification. The data were quantified with imageJ and then presented as mean % CMAC ± SD/10 fields/experiment (n = 3). M) (1) CDH2–Cx43 complex in the cytosol shown moving to the cell membrane where GJIC is established with BM niche cells. (2) CDH2 knockdown prevents Cx43 membrane localization, blunting CSC to form GJIC with BM niche cells. (3) Cx43 knockdown decreases GJIC on the membrane.
Source data are available for this figure.

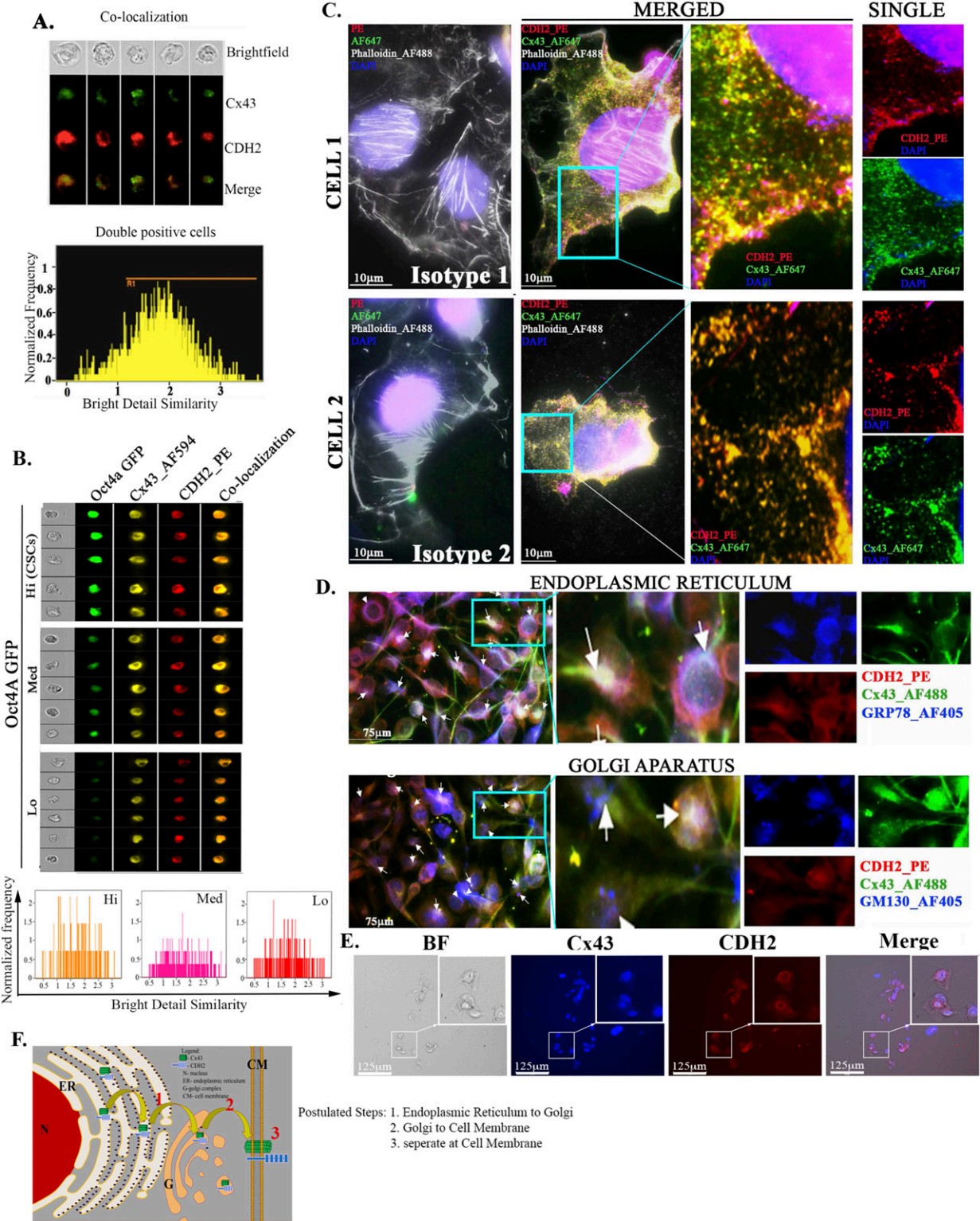

**Figure 2  Colocalization of CDH2 and Cx43 in unsorted BC cells.**
**(A)** ImageStream analyses of single cells for colocalized CDH2 (PE) and Cx43 (AF610) in MDA-MB-231. Upper panel: top row, bright-field cells; lower rows, fluorescence imaging of single cells; Lower panel: histogram of fluorescence distribution for the merged images. **(A, B)** Upper panel: ImageStream analyses was repeated as for "(A)," except with different BC cell subsets (Oct4[hi], Oct[med], and Oct4[lo]) from MDA-MB-231-Oct4a-GFP cells; lower panel: histogram of fluorescence distribution for the merged images. **(C)** Nanoscale imaging of MDA-MB-231, labeled with anti–CDH2-PE (red) and anti–Cx43-AF647 (green), actin with AF488 Phalloidin (white/grey) and nuclei with DAPI (Blue). Slides were imaged with the Nanoimager S Mark II, 1,000× magnification. Cells 1 and 2 represent images of colocalized CDH2 and Cx43 (yellow). Enlarged

sites (Fig 3C, colored regions). Such interaction appeared to be important for molecular signaling, based on STRING, which predicted the incorporation of key proteins, linked to dormancy such as tight junction protein 1 and β-catenin (CTNNB1) (Fig 3D) (Kim et al, 2017; Paquet-Fifield et al, 2018).

Next, we used protein extracts from unsorted BCCs to immunoprecipitate Cx43 with anti-CDH2 and noted a faint band (Fig 3E). Similar immunoprecipitation with anti-Cx43 showed bright band for CHD2 (Fig 3F). In addition, immunoprecipitation with extracts from BCCs with ectopic CDH2-Flag or Cx43-HA, using anti-Flag showed a strong band for HA, confirming CDH2-Cx43 interaction (Fig 3G). Whole cell extracts from CSCs, when immunoprecipitated with anti-Cx43, resulted a bright band for CDH2, indicating efficient CDH2-Cx43 interaction in CSCs as compared with unsorted BCCs (Fig 3H). This latter observation is consistent with the imaging studies (Fig 2).

To validate CDH2-Cx43 interaction further, we randomly selected BM biopsy from a patient with BC (Patient 3) for proximity ligation assay (PLA) (Fig 3I). The PLA probe would amplify only if CDH2 and Cx43 are <40 nm apart, which implies direct interaction (Alam, 2018). Indeed, we observed bright amplification in the patient sample whereas no signal was observed with isotype control. This finding combined with the immunoprecipation strongly supports CDH2 and Cx43 interaction.

### CDH2 in CSC dormancy within the bone marrow

We validated the in vitro findings showing a role for CDH2 in BC dormancy using immune deficient mice. CSCs from the scramble and CDH2/Cx43 knockdown MDA-MB-231 were isolated as explained in Fig 1E and then injected intravenously into the tail vein (Fig 4A). At day 10, postinjection, cells from mice femurs were analyzed for human and mouse GAPDH by real-time PCR. The results, human/total (mouse + human) GAPDH indicated significantly ($P < 0.05$) more human cells with scrambled shRNA as compared with CDH2 or Cx43 knockdown cells, indicating reduced BCCs in femurs (Fig 4B lower graph). The top panel (Fig 4B) showed baseline dormancy femurs of mice injected with CSCs carrying scramble shRNA–red fluorescence indicates cytokeratin+ cells; white arrows indicate relatively few Ki67+ cells (red/pan-cytokeratin + blue/Ki67 = purple/pink).

As compared to scramble shRNA, CDH2 knockdown led to reduced human cells in mouse femurs (Fig 4C, last row). Lungs, brain, and liver are common metastatic site for human BC (Boire et al, 2019). We therefore imaged sections from these tissues for metastatic BCCs. The number of CSCs (yellow = GFP/CSCs + RFP/shRNA) in the lungs and brain were increased when CDH2 or Cx43 was knockdown, relative to scramble shRNA (Fig 4C, top two rows, yellow representing CSCs). We did not include the image for liver because of undetectable BCCs, regardless of the injected BCCs. We quantified the data from images of six mice, 10 fields/mouse. There was a significant ($P < 0.05$) decrease in CSCs in the femurs as compared with scramble shRNA (Fig 4D). Furthermore, as the number of CSCs

decrease in femurs, we observed increases in lungs and brain (Fig 4D). The CSCs entering the lungs and brain showed increased Ki67, indicating enhanced proliferation (Fig S4A).

M2 MΦs, a component of BM stroma, facilitate BC dormancy by establishing GJIC with CSCs (Walker et al, 2019). Because CDH2 knockdown CSCs retained Cx43, we asked if transplantation of M2 MΦs in the non-obese diabetic (NOD) scid gamma mouse (NSG) immunedeficient mice would be able to reverse the ability of CDH2 knockdown CSCs to adapt dormancy in femurs. We injected the M2 MΦs with cell tracker blue to track their entry into BM (Fig S4B). Although M2 MΦs enter the femurs, they did not change the migratory pattern of the CDH2 knockdown cells (Fig S4B and C). In summary, we showed that CDH2 knockdown CSCs lose preference for the BM.

### RNA-Seq analyses of CDH2 knockdown CSCs

We dissected the complex role of CDH2 in BC dormancy by subjecting the CSCs, knockdown for CDH2 or Cx43 or scramble shRNA to RNA-Seq. Principal component analysis indicated clustering among the groups (Fig 5A). There were 956 and 368 differentially expressed genes in Cx43 and CDH2 knockdown versus scramble shRNA, respectively (Fig S5A and B). Hierarchical clustering showed distinct gene populations among the groups (Fig 5B). We selected the overlapped genes between Cx43 and CDH2 knockdowns to determine the common function when these two genes were knockdown. IPA analyses indicated up-regulated pathways with links to DNA metabolism and apoptosis (Fig 5B/grey box). The output also indicated increases in pathways relevant to metastasis, for example, EMT regulation, cytokine (CXCR4 and TGFβ) signaling, cell cycle regulation, and inflammation (Fig 5C and D).

Gene set enrichment analyses (GSEA) indicated up-regulation of G2M checkpoint and E2F targets in the knockdown cells, consistent with increased cell proliferation (Figs 5E and S5C). The genes decreased in both knockdown cells included those linked to EMT, such as Zeb1, retinoblastoma (Rb), stem cell–associated and p53 (Figs 5F and S5D). Because CSCs express EMT proteins (Mani et al, 2008), we verified these changes by Western blot with extracts from BCCs, unsorted, knockdown for CDH2 or Cx43, or scramble shRNA. There were negligible changes in the expression of metastatic-related genes—vimentin, twist, slug, β-catenin, and Zeb1 (Fig 5G). We follow up on Zeb 1 because of its role in maintaining CSCs (Pérez et al, 2021). This was addressed with CSC extracts, resulting in a faint double band in the scramble and knockdown cells (Fig 5H). Similar analyses with CSC extracts indicated deceased bands for vimentin, β-catenin, and twist in Cx43 knockdown cells without any change with CDH2 knockdown (Fig 5H). MDA-MB-231 is a mesenchymal type cells and express high levels of CDH2. Despite its knockdown, E-cadherin remained undetectable (Fig 5H).

MAPK pathway has been reported to regulate stem cell renewal (Niu et al, 2017). We therefore assessed proteins associated with this

---

sections, second from right column, with the respective channels at the far-right column. Isotype control is shown at the far-left column. **(D)** MDA-MB-231 cells were colabeled with anti–CDH2-PE, anti–Cx43-AF488, and anti–GRP78-AF405 (ER) or anti–GM130 AF405 (Golgi G). The cells were imaged with EVOS FL2 Auto at 200× magnification. **(E)** MDA-MB-231 cells were treated with 5 µg/ml brefeldin-A. After 24 h, the cells were washed and then colabeled with anti–CDH2-PE and anti–Cx43-AF405 followed by imaging with EVOS FL2 Auto at 200× magnification. **(F)** Cartoon summaries the organelle labeling for colocalized CDH2 and Cx43 from ER to G and then to cell membrane.

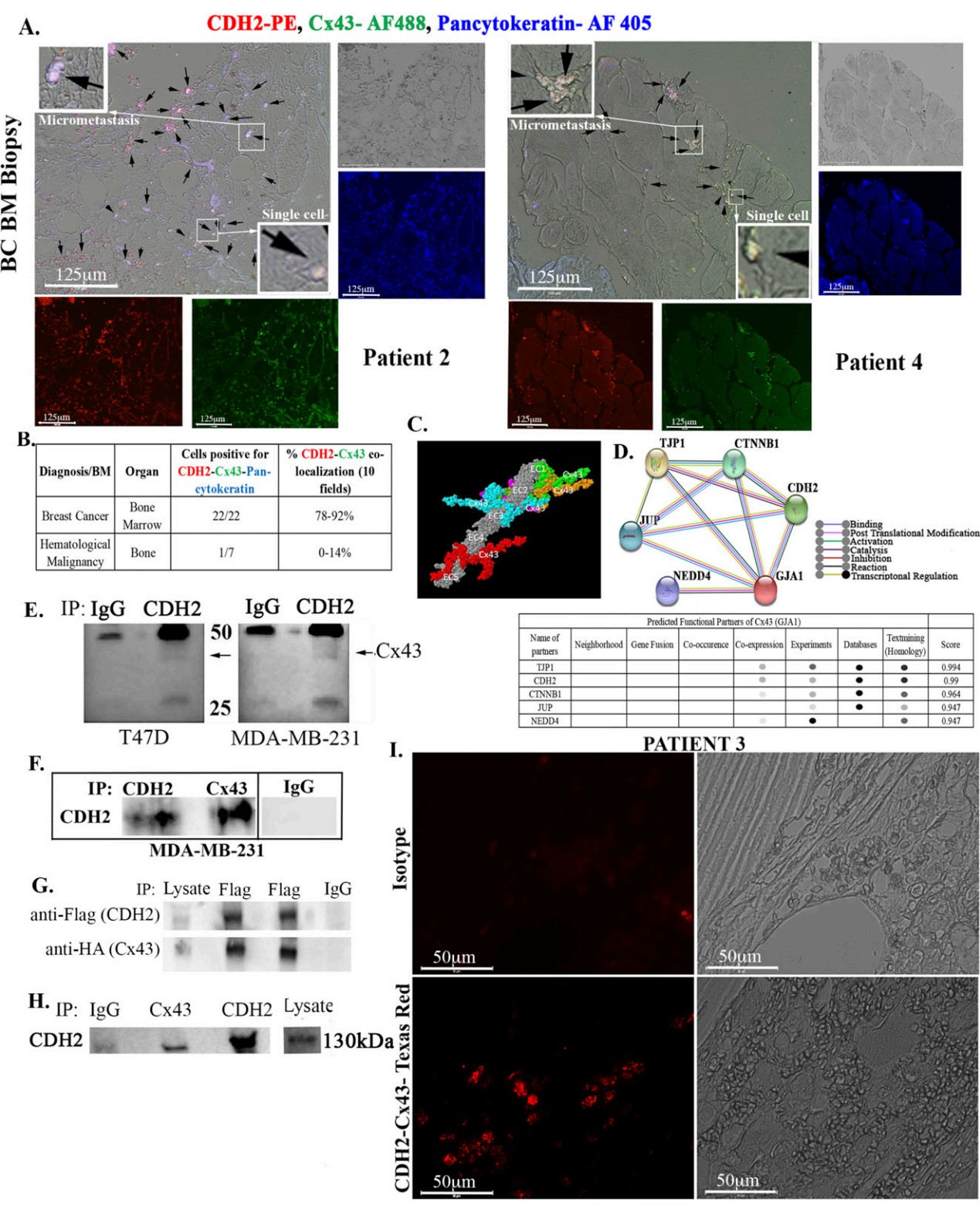

**Figure 3. CDH2-Cx43 interaction and colocalization in BM biopsies of BC patients.**

**(A)** Representative images of tissue sections from biopsies of BC patients (A and S2), hematological malignancy (S3), and benign tumor (S3). The slides were colabeled with anti–CDH2-PE (red), anti–Cx43-AF488 (green), and anti–pan-cytokeratin-AF405 (blue). Images were acquired with EVOS FL Auto 2, 200× magnification. Black arrows show colocalized Cx43 and CDH2 in the white areas (red + green + blue). Inset, zoomed regions of colocalized proteins. **(B)** Table summarizes the total number of sections positive for colocalized CDH2, Cx43 in pan-cytokeratin + cells. ImageJ software was used to count the colocalized cells in 10 fields/slide. The last column shows the percentages of colocalized CDH2-Cx43. See also Table S1 and Figs S2 and S3. **(C, D)** Computational and functional prediction of CDH2–Cx43 interaction using ZDOCK (C) and

pathway in the knockdown cells and scramble CSCs. Western blot showed similar bands for p38 between knockdown and scramble CSCs from MDA-MB-231 (Fig 5I). p-Erk1/2, which is associated with proliferation in the MAPK pathway, was increased the scramble shRNA CSCs (Fig 5I) (Mebratu & Tesfaigzi, 2009). Together, the data indicated that CDH2 knockdown leads to the loss of stemness with the potential for CSCs to become sensitive to apoptosis and to be more proliferative. Our conclusion is based on the following: increases in genes linked to apoptosis and DNA metabolism (Fig 5B, grey region), down-regulated stem cell—related pathways (Fig 5F), decreased p-ERK, and increased pathways to propagate cell cycle.

## Increased proliferation and reduced stemness of CSCs in CDH2 knockdown

The RNA-Seq data and follow-up Western blots suggested that deficient CDH2 or Cx43 in CSCs might lead to loss of stemness, acquired proliferative capacity, and possible sensitivity to apoptosis. These points were deduced from the top pathways associated with stemness, for example, mitochondrial dysfunction (Figs 5D and S5C and D). Therefore, we performed functional studies to validate these findings. First, we determined if there is reduced stemness by flow cytometry for GFP in the knockdown CSCs. Because the CSCs expressed p-Oct4a-GFP, GFP intensity served as a surrogate of the relative cell maturity (Patel et al, 2012; Bliss et al, 2018). Cx43 and CDH2 knockdown significantly ($P < 0.05$) reduced primitive BCCs (Oct4a[hi] and Oct4a[med]) and concomitantly increased Oct4a[neg] BCCs, suggesting differentiation with CDH2 and Cx43 knockdown (Fig 5J, relative cell maturity within the diagram, Fig S5G and H). These phenotypic analyses correlated with reduced ability of CDH2 or Cx43 knockdown CSCs to form tumorspheres (Fig 5J spheres).

The loss of stemness in the knockdown CSCs, combined with increased expression of genes involved in cellular proliferation led us to study the relative migration of CSCs, knockdown for Cx43 or CDH2, or transfected with scramble shRNA. Scratch assay showed a significant ($P < 0.05$) increase in cell movement at 50 h, relative to scramble shRNA, indicating increased migration by the knockdown cells (Fig 5K and L). These results are consistent with the RNA-Seq data pointing to increased cell cycle and DNA metabolism in the CDH2 or Cx43 knockdown BCCs (Figs 6A–F and S5E and F and Tables S2 and S3). Western blot analysis confirmed increase levels of proteins linked to cell cycling progression (Fig 6G). Indeed, cell cycle analyses indicated ~50% less of low RNA in the knockdown cells as compared with scramble shRNA (Fig 6H and I). Further analyses using a reporter gene system under the control of cyclin D1 5' regulatory region showed a significant ($P < 0.05$) decrease in luciferase activity in CDH2 and Cx43 knockdown BCCs, relative to scrambled shRNA (Fig 6J). Because cyclin D1 is required for G0/G1

transition, flow cytometric analyses in cells labeled with 7-AAD showed an increase in the S-phase in the knockdown cells (Fig 6K). In summary, CDH2/Cx43 knockdown caused BCCs to cycle, as noted with the cells predominantly in the S-phase.

## Increased apoptotic pathways in CDH2 knockdown CSCs

There were increased apoptotic genes in the CDH2 and Cx43 knockdown CSCs (Fig 5). Further analyses for apoptotic genes in the RNA-Seq data verified their increases in CDH2 knockdown CSCs, relative to scrambled shRNA (Fig 7A). We corroborated these studies in Western blot for caspase 3 and 7 using protein extracts from CDH2 knockdown CSCs (Fig 7B, normalized band densities at right). Functional analyses for caspase activity indicated a significant ($P < 0.05$) increase in CDH2/Cx43 knockdown CSCs, relative to scramble shRNA (Fig 7C). Caspase activity was further increased when we treated the knockdown CSCs with 220 μg/ml carboplatin (Fig 7C). We selected this chemotherapy because this allowed us to use the same treatment for both triple negative and positive BCCs. The efficiency of chemotherapy varied because similar studies with 1 μM doxorubicin resulted in reduced caspase activity as compared with carboplatin (Fig 7C). The sensitivity of the knockdown cells was in line with reduced P-gp (Fig 7D). These findings correlated with increased cell death ($P < 0.05$) with 220 μg/ml carboplatin or 1 μM doxorubicin treatment, relative to scramble shRNA (Fig 7E and F). By 96 h, the treated cells showed negligible surviving CDH2 and Cx43 knockdown CSCs whereas similar studies with scramble shRNA resulted in negligible cell death (Fig 7E/zoomed section, Fig 7F).

We asked if the in vitro sensitivity of the knockdown CSCs to chemotherapy also occurs in vivo. We therefore injected NSG mice intravenously with $5 \times 10^5$ CSCs, knockdown for Cx43 or CDH2, or scrambled shRNA and then treated twice with carboplatin (5 mg/kg) (Fig 7G). Sections of decalcified paraffin femurs were labeled with anti–pan-cytokeratin (Texas Red) and anti-Ki67 (Blue). Carboplatin reduced the number of BCCs in femurs when the mice were injected with CDH2 and Cx43 knockdown CSCs, as compared with vehicle (Fig 7H and I). BCCs, enumerated in 10 fields/femur from five mice ± SD, indicated a significantly ($P < 0.05$) reduced BCCs relative to scramble shRNA (Fig 7I). Carboplatin treatment cleared CDH2/Cx43 knockdown CSCs in the lungs of mice, as indicated by a lack of yellow cells (GFP + RFP) (Fig S4D). This contrasted increased number of BCCs in the lungs and brain and decreased BCCs in femurs when mice, injected with CDH2/Cx43 knockdown CSCs, were untreated (Fig 4). In summary, CDH2 knockdown increased apoptotic pathways, independent of p53 (Fig 5D), favoring responses to carboplatin (Fig 7J).

---

STRING (D), respectively. CDH2, Cadherin-2 (N-cadherin); TJP1, tight junction protein 1; CTNNB1, β catenin; GJA1, gap junction alpha protein 1(Cx43); JUP, junction plakoglobin; NEDD4, neural precursor cell expressed, developmentally down-regulated 4. **(E)** Whole cell lysates from MDA-MB-231 and T47D were immunoprecipitated with anti-CDH2 or IgG and then electrophoresed on 12% SDS–PAGE. The membranes were blotted with anti-Cx43 (light band at 43 kD). **(F)** Whole-cell lysated from MDA-MB-231 cells were subjected to immunoprecipitation (IP) with anti-CDH2 or anti-Cx43. The membrane was blotted with anti-CDH2. **(G)** MDA-MB-231 was transfected with pCMV2-CDH2 Flag and pcDNA 3.2-Cx43-HA. Protein lysates were immunoprecipitated with anti-Flag or anti-IgG and blotted with anti-Flag or anti-HA. **(H)** Lysates from cancer stem cells, isolated from MDA-MB-231 were immunoprecipitated with anti-IgG, anti-Cx43, or anti-CDH2. The samples were electrophoresed and then blotted with anti-CDH2. **(I)** BM biopsy from BC patient (#3) was subjected Proximity Ligation Assay with anti-CDH2 and anti-Cx43. The proximity of the two antibodies was determined using EVOS FL Auto 2, 600× magnification. Control slide was labeled with isotype control.

Source data are available for this figure.

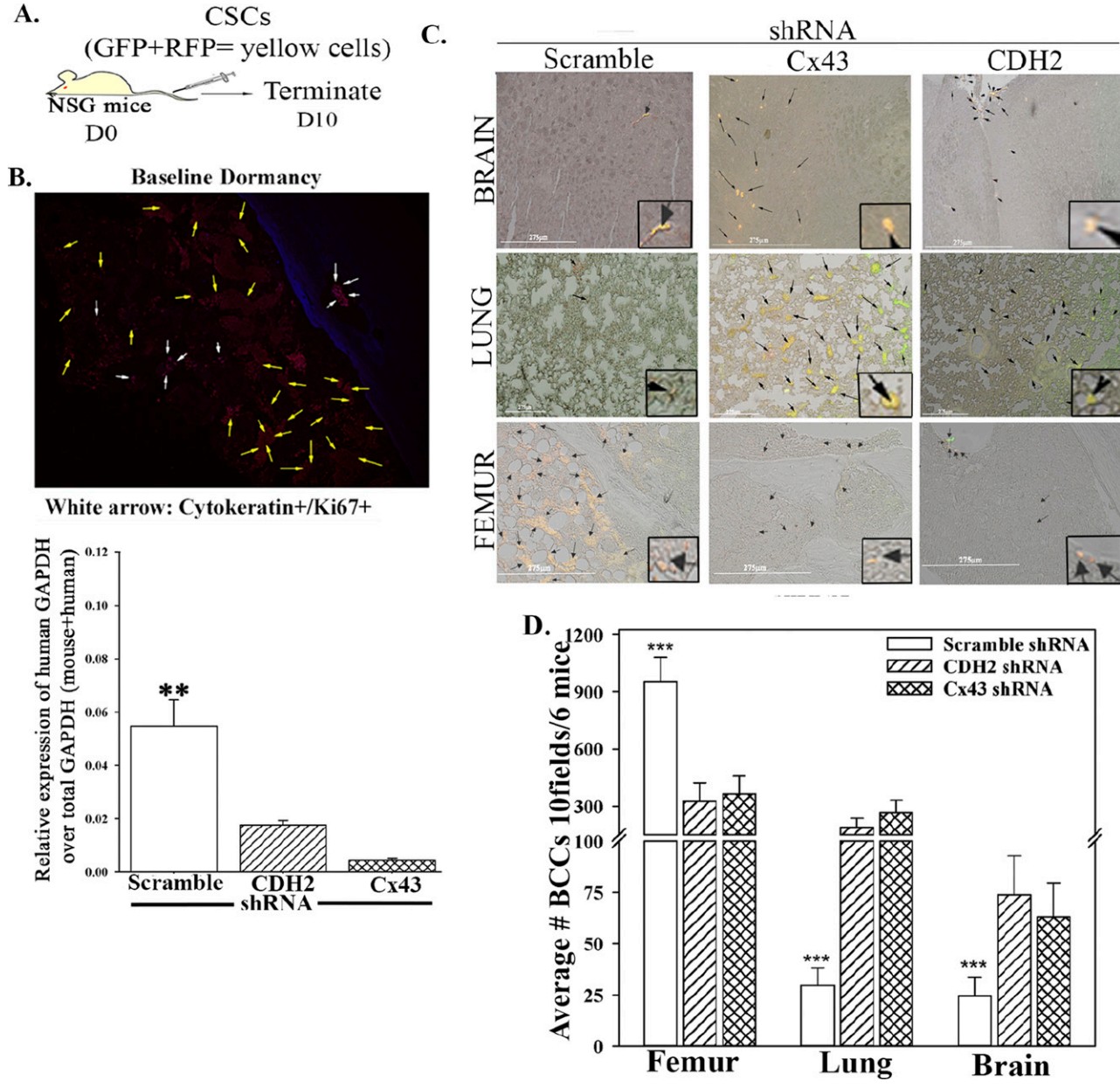

**Figure 4. Organ localization of BC cells knockdown for CDH2, in vivo.**
**(A)** Cartoon summarizes study design. **(B)** 5 × 10⁵ cancer stem cells from MDA-MB-231-Oct4-GFP, knockdown for CDH2 (RFP) or Cx43 (RFP), or control with scramble shRNA (RFP) were injected intravenously into NSG mice. At day 10, the lungs, brain, and femurs were harvested. Cells from one femur/mouse were used to isolate total RNA for real time PCR with primers for human and mouse GAPDH. The results are presented as human GAPDH/Total GAPDH (human & murine) ±SD, n = 6. **P < 0.05 versus CDH2 and Cx43 knockdown. Fluorescence image: yellow arrow represents pan-cytokeratin positive cells (Texas red); white arrows: Ki67-Alexa-AF405–positive cycling cells (blue + red = pink). **(B, C)** Paraffin embedded from "(B)" were imaged for cancer stem cells (RFP for knockdown vector + GFP for Oct4A = yellow) with EVOS FL Auto 2. Shown are 200× magnification for each image, representing six mice/group. **(D, E)** Quantitation by ImageJ of BC cells in the femur, lungs, and brain from "(E)," mean of 10 fields/slide for six mice ± SD. ***P < 0.05 versus lungs or brain.

# Discussion

We report on the mechanisms by which CDH2 facilitates GJIC between CSCs and BM stromal cells. CDH2 and its associated pathways are responsible for Cx43-mediated GJIC, maintaining stemness and drug resistance (Figs 5–7). There is a great necessity to reverse BC dormancy to achieve chemosensitivity and to prevent BCCs from entering

dormancy. Targeting dormant BCCs has been a clinical challenge due to the dormant BCCs showing properties of CSCs (Carcereri de Prati et al, 2017; De Angelis et al, 2019). Targeting the multipotent capacity of CSCs will likely affect HSCs, resulting in BM toxicity. Similarly, Cx43, which is responsible for GJIC between CSCs and BM niche cells, is also required for normal hematopoiesis (Patel et al, 2012; Taniguchi Ishikawa et al, 2012). This study provides newly elucidated pathways to reverse BC dormancy

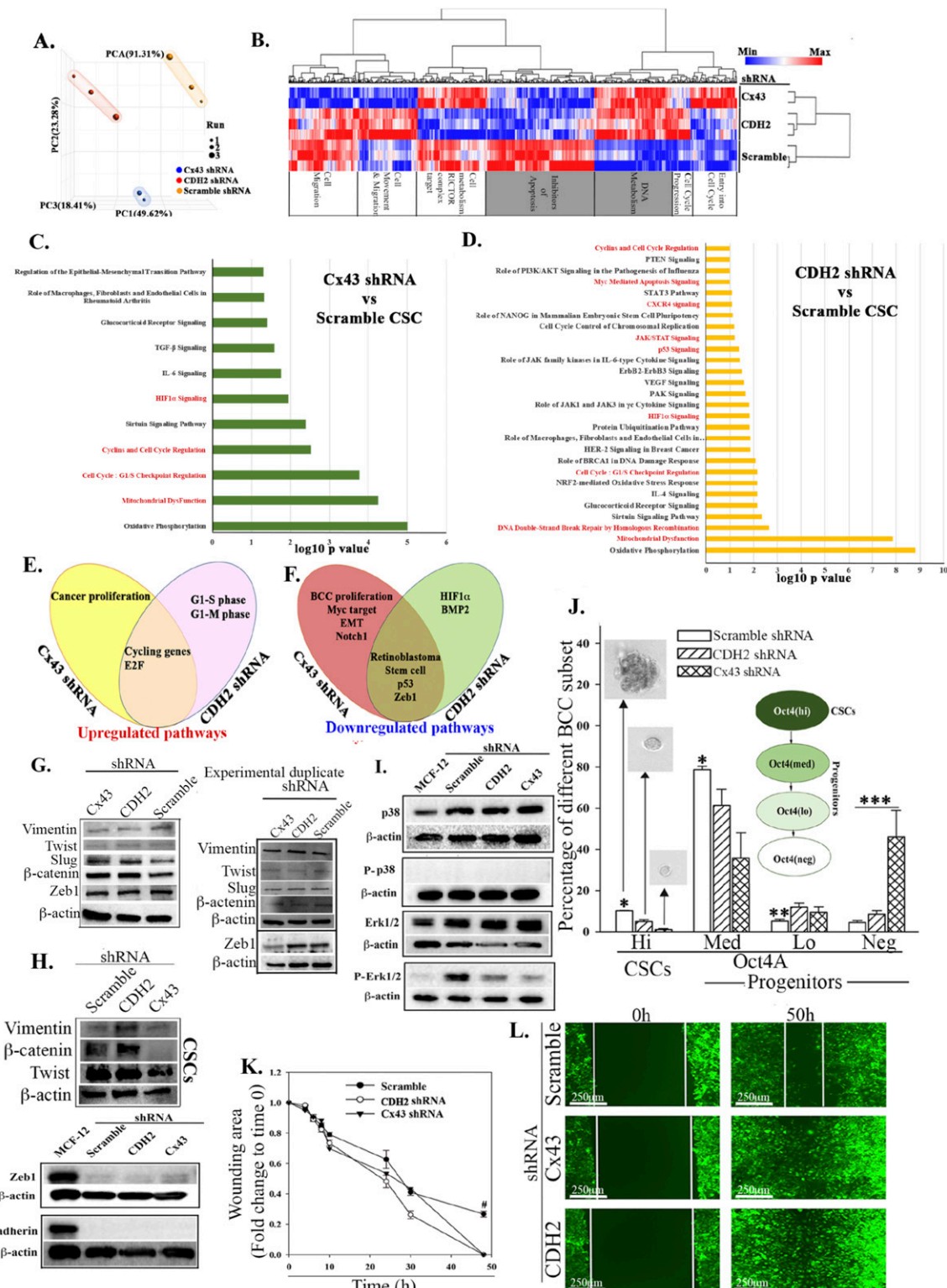

**Figure 5  Next generation sequencing of cancer stem cells (CSCs), knockdown for CDH2 and Cx43.**
**(A)** Principle component analyses of RNA-Seq data from the CSCs of the knockdown and control (scrambled) cells. **(B)** Heat map with genes using false discovery rate <0.05 and fold change for CDH2 versus scramble of 1.66 and Cx43 versus scramble 1.5. **(C, D)** IPA analyses showing significant changes, Cx43 or CDH2 knockdown versus scramble shRNA. The pathways were ranked using significance score ($P < 0.01$), also known as $-\log_{10}$ $P$-value times activation z-score. Red highlights: relevance to cell cycle and stem cell maintenance. **(E, F)** Venn diagram showing up- and down-regulated pathways of Cx43 or CDH2 knockdown versus scramble using significance scores of q < 0.05. **(G, H)** Western blot for epithelial mesenchymal transition proteins with whole cell extracts from the following: unsorted MDA-MB-231 (G); CDH2 or Cx43 knockdown

by rendering CSCs chemosensitive (Fig 8). Indeed, targeting CDH2 and its key pathways trigger cell proliferation, loss of stemness and activated pro-apoptotic pathways to chemosensitize resistant CSCs (Figs 4–7).

Knockdown of CDH2 and Cx43 led to reduced CSCs in femurs but concomitant increased number of BCCs in the brain and lung (Fig 4C). These findings strongly suggested that similar loss of stemness in CSCs within femurs could reverse dormancy. The experimental studies indicated that the CDH2 knockdown cells entering the brain and lungs were chemosensitive, based on the clearance of BCCs in these organs when the mice were treated (Fig S4D and 7H and I). These findings are relevant to treatment as the data provide information to combine CDH2 inhibitor such as a γ-secretase inhibitor with chemotherapy. Such combination will ensure targeting of the CSCs without allowing them to home to tertiary sites. CDH2 was high in T47D but Cx43 was increased in MDA-MB-231 (Fig 1A). Despite these differences, CSCs from both cell lines show preference for GJIC with BM microenvironmental cells (Patel et al, 2012), indicating relevance to triple-negative and positive BCCs. This study emphasized on MDA-MB-231 because of poor prognosis associated with triple-negative BCCs across the metropolitan areas, including our inner city medial school oncological facility.

The loss- and gain-of-function studies on CDH2 in GJIC between CSCs and BM niche cells (Fig 1) indicate a central role for CDH2 that explains our finding on Cx43-mediated gap junction (Patel et al, 2012). The studies did not address if deficient CDH2 could be responsible for CSC exit from the BM, but we determined that its decrease caused migration of CSCs from the blood into brains and lungs. Because another report showed preference of CSCs for BM within 72 h, we deduced that CDH2 may have another role in directing CSCs to BM (Patel et al, 2012). Going forward, it is important to track the timeline movement of CDH2-deficient CSCs from BM to other organs. It is also important to identify the interacting sites between CDH2 and Cx43, including orthogonal techniques to quantitate CDH2–Cx43 interaction in different organelle. Such studies will provide insights on the 3-dimensional structure and perhaps the effects on linked cell signaling pathways. Our in silico analyses show predicted interaction sites, which were validated by different techniques, including PLA and nanoimaging to show Cx43-CDH2 colocalization (Fig 3). Thus, rather than inhibiting CDH2 alone, an exciting follow-up would be to focus on mapping the CDH2 motifs that are specifically involved in the interaction with Cx43, which is up-regulated in CSCs. This would allow for the development of pharmacological agents to block CDH2-Cx43 interactions in dormant chemoresistant CSCs.

The colocalized Cx43 and CDH2 in cancer cell lines were similar in 22 BM biopsies from BC patients. Interestingly, there was colocalized Cx43/CDH2 (within very few cells) in BM biopsies of patients with hematological malignancy. One slide from hematological malignancy was positive for cytokeratin, suggesting an undiagnosed epithelial malignancy. Furthermore, using PLA with BM biopsy, we not that the axis between CDH2 and Cx43 is not limited to colocalization; instead these two molecules for a complex with each other (Fig 3I). The data from patients, although limited, are exciting and warrant further analyses. Nonetheless, the data support the idea that CDH2–Cx43 interaction could be required for establishing and plausibly maintaining malignancy in BC patients. Because BC dormancy can occur at any time during the disease and after treatment, the continued colocalization between Cx43 and CDH2 in posttreatment samples confirmed the importance of such colocalization during dormancy.

The question is why there is a preferential effect of tumor cells to BM? This is a clinical observation with no clear answer. The CSCs with higher CDH2 and Cx43 quickly home to the BM whereas cells with lower levels of these molecules remain at other sites such as lungs. It is unclear if CDH2 and Cx43 guide CSCs to BM. We now show that these molecules facilitate the cells' survival in BM and possibly during long-term dormancy. Thus, this study begins to advance our understanding on the mechanisms by which BM niche cells maintain BC dormancy with the CDH2-Cx43 axis as a central theme for further studies. This would lead to the discovery of new mechanisms to reverse and prevent BC dormancy for therapeutic targeting. Interestingly, M2 macrophages (MΦ), shown to mediate BC dormancy (Walker et al, 2019), could not replace CDH2 knockdown in CSCs with respect to re-establishing dormancy in BM (Fig S4). Because M2 MΦ can also form GJIC with BM niche cells, it is possible that the loss of CDH2 in CSCs also prevented dormancy with MΦ. The studies with M2 MΦ taught us that perhaps the secretome from M2 MΦs might be less important and CDH2 warrants further studies.

The increased cell proliferation with enhanced migration of BCCs in mouse brain and lungs following CHH2 knockdown resembles reverse dormancy of BCCs when the dormancy cells become metastatic. We did not observe any change in EMT proteins with decreased stemness in the CDH2 knockdown CSCs. This was not a surprise because EMT is not a decisive factor of stemness (Kalluri & Weinberg, 2009; Mitra et al, 2015). The identified pathways in this study provide new avenues for the development of pharmacological treatments to reverse BC dormancy without HSC toxicity. These findings could also serve as a platform for preclinical studies, perhaps by repurposing drugs. BM toxicity analyses are required because there are mixed information on the literature on the role for CDH2 in HCS maintenance (Kiel et al, 2009; Arai et al, 2012).

We immunoprecipitated CDH2 in MSCs and also showed the interaction with Cx43 in MSCs (Fig S1D). This finding is significant because MSCs, which are also found in the perivascular niche (Ehninger & Trumpp, 2011), are important for immediate transition to dormancy (Sandiford et al, 2021). In summary, we carried out knockdown of CDH2 in CSCs and demonstrated reduced cells in femurs. Because we previously demonstrated GJIC formation by CSCs and resident BM cells, the loss of stemness in CDH2

---

CSCs, CSCs with scrambled shRNA (H). **(I)** Western blot for p38, P-p38, Erk1/2, and p-Erk1/2 with whole cell extracts from MCF12A and MDA-MB-231 CSCs with or without CDH2/Cx43 knockdown. **(J)** BC cell subsets were quantitated by flow cytometry, based on GFP with MDA-MB-231–p-Oct4a-GFP, also knockdown for CDH2 or Cx43 or control/scramble shRNA. The cells were gated as for Fig 1B and % subset plotted as mean ± SD, n = 4. Each subset was analyzed for tumorsphere and representative sphere shown for Oct4[hi] and Oct4[med]. No tumorsphere was formed with Oct4[lo or neg]. *$P < 0.05$ versus Oct4[hi] or Oct4[med] knockdowns; **$P < 0.05$ versus knockdowns; ***$P < 0.05$ versus scramble or knockdown. **(K, L)** Scratch assay with CSCs from MDA-MB-231, knockdown for CDH2 or Cx43 or control/scramble shRNA. The scratch areas were imaged at different times with EVOS FL Auto 2. The timeline changes are presented as mean fold changes ± SD, n = 3. Fold change is calculated as experimental/time 0 h. #$P < 0.05$ versus CDH2 and Cx43 knockdown. Scratch image (100×) is shown for the 50 h end point.
Source data are available for this figure.

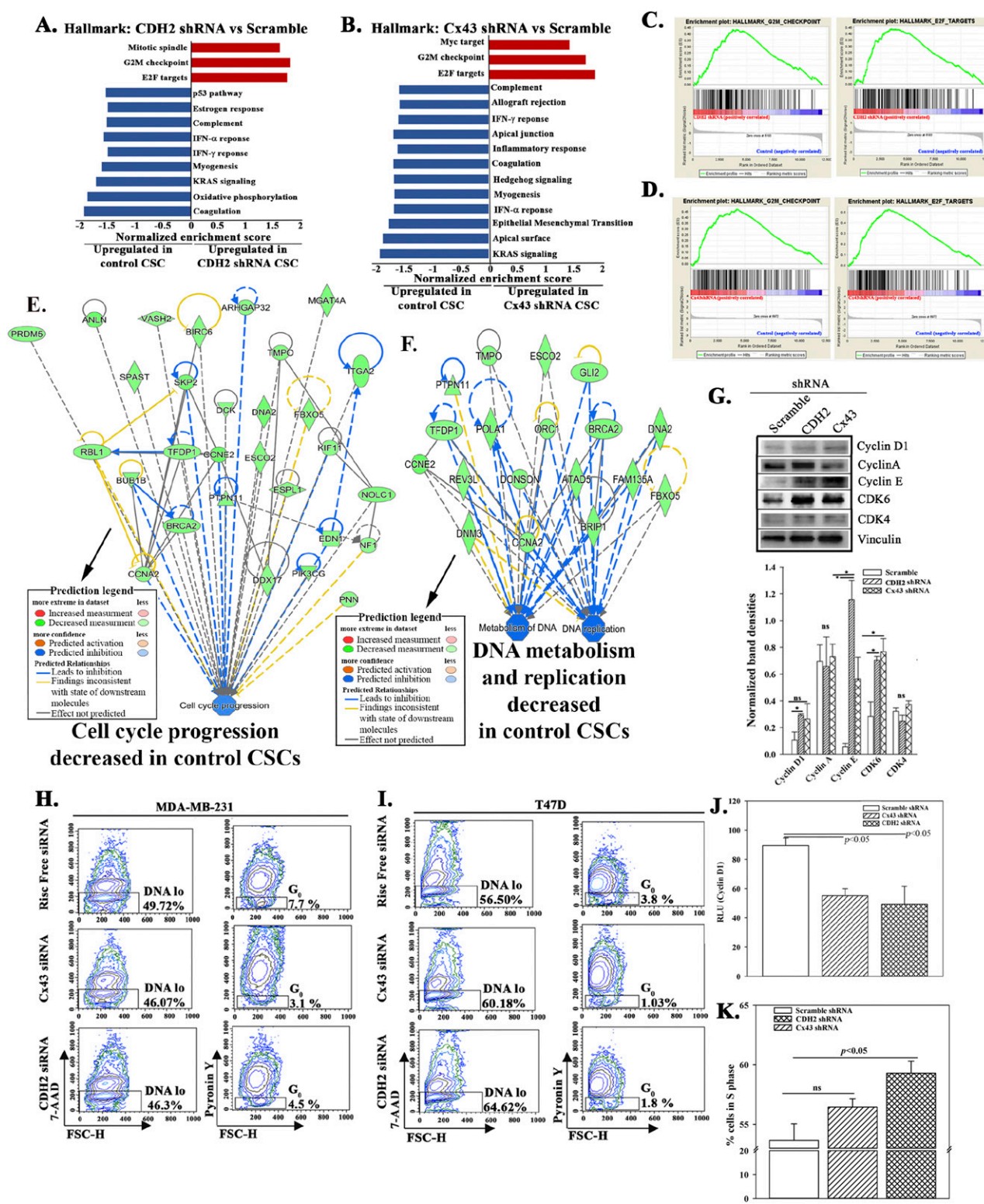

**Figure 6. Cycling analyses of cancer stem cells (CSCs), knockdown for CDH2 or Cx43.**
**(A, B)** Gene set enrichment analyses hallmarks for CDH2 or Cx43 knockdown versus scramble CSCs. The red and blue bars indicate up-regulated hallmarks in CDH2 knockdowns and control/scramble, respectively, with significance score of q < 0.05. **(C, D)** Representative gene set enrichment analyses graph showing top hallmarks, G2M and E2F, which were enriched in CDH2 and Cx43 knockdown, respectively. **(E, F)** IPA output showing cell cycle progression, DNA metabolism, and replication pathways that were down-regulated in CSCs (scramble), relative to CDH2 or Cx43 knockdown cells. Significance score used $P < 0.01$. **(G)** Western blot for cycle proteins with whole extracts from MDA-MB-231, knockdown for CDH2, and Cx43 or transfected with scramble shRNA. **(H, I)** The normalized band densities are shown below, mean ± SD, n = 3.

knockdown led us to deduce a reduction of GJIC. Based on the overall data, it appears that CDH2 knockdown transitioned the CSCs into increase in metastatic cycling cells/chemosensitive cells (Fig S4A). CDH2 knockdown CSCs lost their ability to self-renew and show increased apoptotic activity (Fig 7). The molecular changes within the CDH2 knockdown CSCs cause chemosensitization, indicating potential for therapeutic intervention. An interesting finding is the noted brain metastasis in both Cx43 and CDH2 knockdown CSCs (Fig 4C and D). In brain, we observed increase in Ki67 in the BCCs, which is in line with other studies reporting on inflammatory-mediated propagation of brain metastasis (Chen et al, 2016). The involvement of cytokines is supported by the RNA-Seq analyses, which showed cytokine and STAT1 signaling pathways in Cx43 and CDH2 knockdown CSCs, respectively (Fig 5C and D). Overall, our data show that CDH2-dependent pathways involve interaction with Cx43, and the relationship between Cx43 and CDH2, are complex (Fig 8). As an example, the literature reported on a fragment of Cx43 acting as a transcription factor to regulate CDH2 (Kotini et al, 2018). This relationship is underscored by the RNA-Seq data showing both genes regulating common pathways as well as decrease of one gene when the other is knockdown.

# Materials and Methods

## Human subjects

Rutgers Institutional Review Board, Newark, NJ, approved the use of paraffin sections from BM biopsy of BC patients, other hematological malignancy, and benign sections of breast (Table S1). The IRB also approved the use of BM aspirates and peripheral blood from healthy donors (18–35 y).

## Reagents

DMEM, RPMI-1640, L-glutamine, penicillin–streptomycin, FCS, Ficoll–Hypaque, $\beta$-mercaptoethanol, PBS, doxycycline, hydrocortisone, polybrene, sodium chloride, magnesium chloride, NP-40, Triton X-100, bovine serum albumin, protease inhibitor, phosphatase inhibitor, PureProteome Protein A Magnetic Bead, N,N,N′,N′-tetramethylethylenediamine (TEMED), ammonium persulfate, polyvinylidene difluoride membranes, 1-octonal, pyronin Y (PyY), Duolink in situ detection reagent red, and Duolink probe maker were purchased from Millipore Sigma; Tris, Tris–HCl, Alexa Fluor 488 Phalloidin, and hyclone donor equine serum from Thermo Fisher Scientific; $\alpha$-MEM, accutase, 0.05% trypsin–EDTA, OptiMEM, Lipofectamine 3000, Lipofectamine RNAiMAX, Genetecin, 7-aminoactinomycin D (7-AAD), Platinum SYBR Green qPCR SuperMix-UDG Kit, DAPI, Cell Tracker Blue CMAC Dye and Cell Tracker Deep Red dye, High-Capacity cDNA Reverse Transcription Kit, SuperSignal West

Femto Maximum Sensitivity Substrate, TRIzol, and glycerol were purchased from Thermo Fisher Scientific; Tet-free FBS, Lenti-X concentrator, In-Fusion HD Cloning Kit, and p24 Rapid Titer Kit from Takara Bio; Bradford protein assay reagent, glycine, 2× loading dye, and sodium dodecyl sulfate from Bio-Rad; TransIT-Lenti transfection reagent from Mirus; luciferase substrate, $\beta$--galactosidase assay kit, CellTiter-Blue Cell Viability Assay Kit and Apo-ONE Homogeneous Caspase 3/7 assay from Promega; acryl/bis solution (30%) 37.5:1 was purchased from VWR; brefeldin-A from InvivoGen and RNeasy Mini Kit, and plasmid miniprep kit from QIAGEN. Carboplatin and doxorubicin were obtained from university hospital pharmacy.

## Antibodies and cytokines

Mouse anti-human CDK4 mAb (1/1,000 dilution), anti-human CDK6 mAb (1/1,000 dilution), mouse anti-human cyclin E mAb (1/1,000 dilution), rabbit anti-human cyclin D1 mAb (1/1,000 dilution), rabbit anti-human Cx43 (1/1,000 dilution), rabbit anti-human Vimentin (1/1,000 dilution), rabbit anti-human Slug (1/1,000 dilution), rabbit anti-human $\beta$-catenin (1/1,000 dilution), rabbit anti-human Zeb1 (1/1,000 dilution), rabbit anti-human caspase 3 (1/1,000 dilution), rabbit anti-human caspase 7 (1/1,000 dilution), rabbit anti-human MDR1 (1/1,000 dilution), rabbit anti-human E-cadherin (1/1,000 dilution), rabbit anti-human p38 (1/1,000 dilution), rabbit anti-human p-p38 (1/1,000 dilution), rabbit anti-human Erk1/2 (1/1,000 dilution), rabbit anti-human p-Erk1/2 (1/1,000 dilution), and normal anti-rabbit immunoglobulin G (IgG) were purchased from Cell Signaling Technology; rabbit polyclonal anti-human cyclin A (1/1,000 dilution), and goat anti-rabbit IgG-TR were purchased from Santa Cruz Biotechnology; rabbit polyclonal anti-human CDH2 (1/1,000 dilution), rabbit polyclonal anti-human Ki67 (1/200 dilution), and rabbit anti-human vinculin mAb (1/1,000 dilution) from Abcam; rabbit anti-human Twist (1/1,000 dilution), mouse anti-human pan-cytokeratin (1/200 dilution), and mouse anti-human $\beta$-actin (1/2,000 dilution) from Millipore Sigma; goat polyclonal anti-rabbit IgG-HRP (1/5,000 dilution), goat polyclonal anti-murine IgG-HRP (1/5,000 dilution), goat anti-rabbit IgG–Alexa Fluor 405 (1/500 dilution), goat anti-mouse IgG–Alexa Fluor 488 (1/500 dilution), goat anti-rabbit IgG-Alexa Fluor 610 (1/500 dilution), goat anti-mouse Alexa 647 IgG (1/500 dilution), rabbit anti-human HA tag (1/1,000 dilution), rabbit anti-human Flag tag (1/1,000 dilution), goat polyclonal anti-rabbit IgG-AP (1/500 dilution), rabbit anti-human GM130 (1/1,000 dilution), rabbit anti-human GRP78 (1/1,000 dilution), and CD14$^+$ Dynabeads Flow kit from Thermo Fisher Scientific; HRP-mouse anti-rabbit IgG light chain–specific from Jackson Immuno Research, murine anti-CD206-PE (1/20 dilution), murine anti-HLA-DR-APC (1/20 dilution), murine anti-human CDH2-PE (1/20 dilution), FITC murine anti-human CD73 (1/20 dilution), PE murine anti-human CD29 (1/20 dilution), murine anti-human CD90⁻perCp/Cyanine 5.5 (1/20 dilution) from BD Biosciences; and recombinant human M-CSF, and IL-4 from R & D Systems.

*$P < 0.05$ (H, I) MDA-MB-231 (H) and T47D (I) were knockdown CDH2 and Cx43 with siRNA. Control cells used Risc free. The cells were labeled with PyY (RNA) and 7AAD (DNA). Cells with low RNA was gated within the low DNA area. The data were analyzed with BD Analyses software. **(J)** CDH2 or Cx43 knockdown MDA-MB-231-Oct4-GFP were transfected with cyclin D1 reporter gene vector. The reporter gene activity (luciferase) was normalized with p$\beta$–gal activity and the values presented as mean relative luminescence unit ± SD, n = 4. **(K)** CDH2 or Cx43 knockdown MDA-MB-231 were labeled with 7AAD. The cells were analyzed for % cells in S-phase for each BC cell subset using ModFit software. The data are presented as mean S phase±SD, n = 3.
Source data are available for this figure.

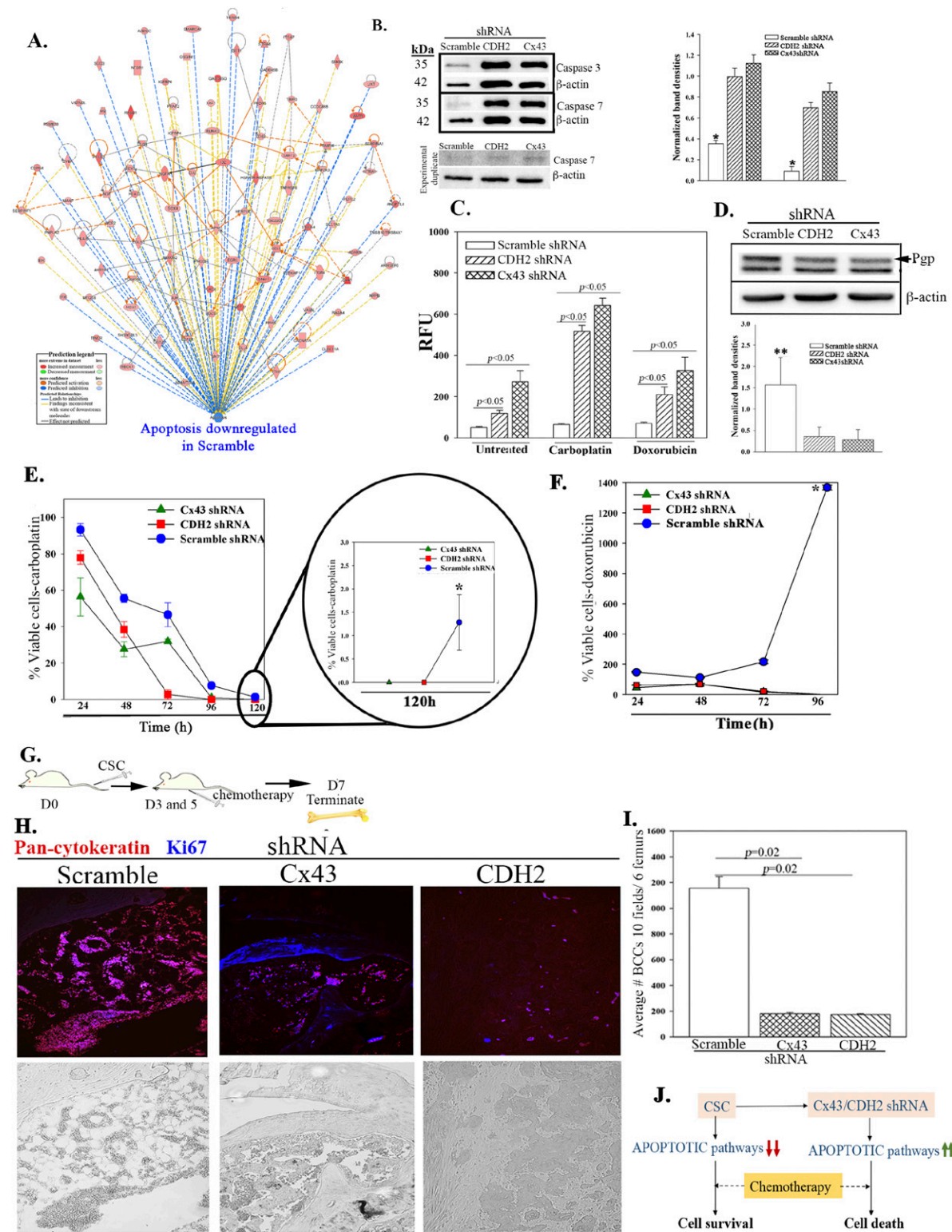

**Figure 7. Apoptotic pathways in CDH2 knockdown cancer stem cells (CSCs).**
**(A)** IPA output shows down-regulation of apoptotic pathways in control (scramble) CSCs relative to CDH2 and Cx43 knockdown CSCs. **(B)** Western blot for caspase 3 and 7 with whole cell extracts from CSCs with scramble shRNA or, CDH2 or Cx43 knockdown. The normalized band densities are shown at right. *P < 0.05 versus CDH2 or Cx43 knockdown. **(C)** Apoptotic activity was analyzed with Apo-ONE Homogeneous caspase 3/7 assay kit and the relative fluorescence unit presented for MDA-MB-231, knockdown for CDH2 or Cx43, or scramble shRNA. In parallel, the assay was performed with cells, treated with carboplatin (220 mg/ml) or doxorubicin (1 mM) for 4 h. **(D)** Western blot for Pgp with whole-cell extracts from CSCs from MDA-MB-231, knockdown for CDH2 or Cx43, or control/scramble shRNA. Bands were normalized with β-actin

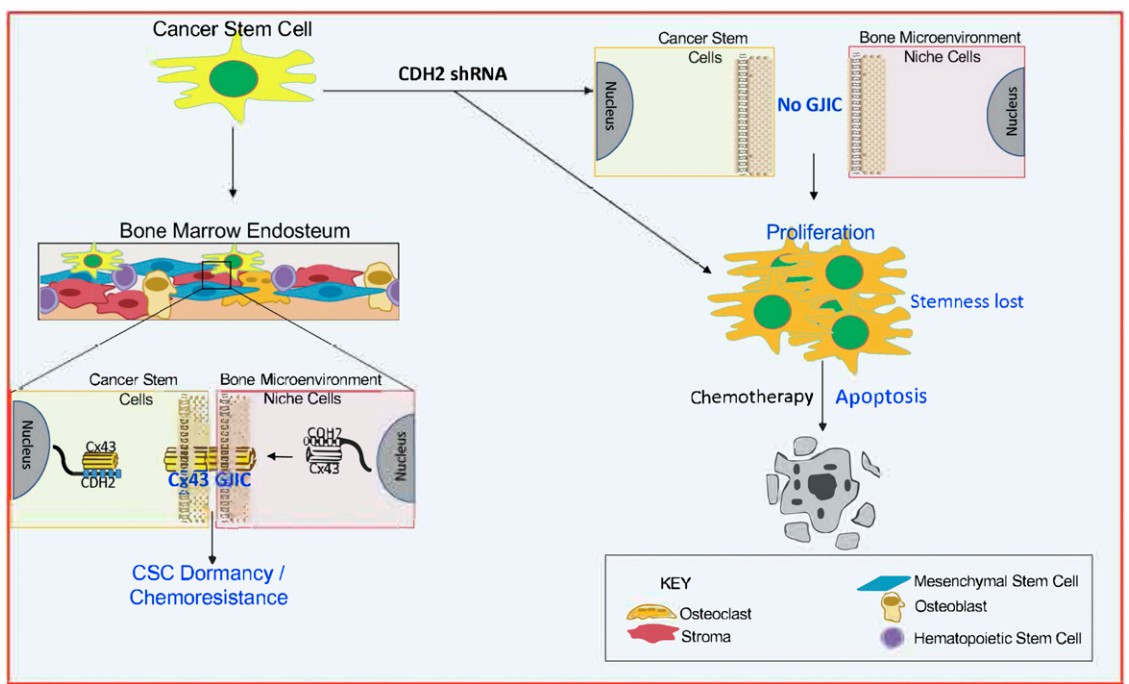

**Figure 8. Summary.**
CDH2 is required for Cx43-mediated GJ between cancer stem cells (CSC) and BM stromal cells. CDH2 knockdown led to increase in apoptotic genes. CDH2 knockdown increase cell cycle of CSCs and metastasis. CDH2/Cx43 knockdown chemosensitized CSCs.

## Vectors and nucleotides

Human CDH2 and Cx43 siRNA and Risc free (control) were purchased from Dharmacon; human cyclin D1 promoter in pGL3-basic (research resource identifiers, RRID:Addgene_32726) and pcDNA 3.2 with Cx43-HA (RRID:Addgene_49851) were obtained from Addgene (which was a donation from Frank McCormick and Anne Brunet laboratory, respectively); pCMV2-CDH2-Flag from Sino Biologicals; and human shRNA pRFP-C-RS with scramble sequence, CDH2-shRNA, or Cx43 shRNA from OriGene Technologies.

Inducible CDH2 and Cx43 expression vectors were created with the lentiviral plasmid pLVX-TetOne-Puro (Takara Bio). The coding region for CDH2 and Cx43 was amplified from pCMV2-CDH2 Flag and pcDNA 3.2-Cx43 HA, respectively, using RT-PCR with following primers: CDH2 forward: 5′-GCA GAG ATC TGG ATC CTC AGT CAT CAC CTC CAC CAT ACA TG-3′, CDH2 reverse: CCC TCG TAA AGA ATT CAT GTG CCG GAT AGC GGG AGC GCTG-3′; Cx43 forward: 5′-CCC TCG TAA AGA ATT CAT GGG TGA CTG GAG CGC C-3′; Cx43 reverse: 5′-GAG GTG GTC TGG ATC CCT AGA TCT CCA GGT CAT CAG GCC-3′. The amplified region was inserted into pLVX-TetOne-Puro using In-Fusion HD Cloning Kit (Takara Bio). The fragment insertion was validated by DNA sequencing at Genewizusing

primer: GGATTAGGCAGTAGCTCTGACGGCCC. The complete vector is hereafter referred as pLVX-CDH2/GS and pLVX-Cx43/GS, respectively.

## Viral propagation, quantitation, and induction

Lentiviral DNA for pLVX-CDH2/GS and pLVX-Cx43/GS was produced with the 3rd Generation Packaging System mix (ABM Inc.). The vectors were co-transfected into HEK293T cells using the packaging mix, lentiviral plasmids, and TransIT-Lenti transfection reagent. The transfection followed the manufacturer's protocol. The transfectants were cultured in 10% DMEM with Tet-free FCS for 48 h. After this, the supernatant containing the virus was collected and filtered through 0.45-µm membrane filtration system. The supernatant was then concentrated using Lenti-X Concentrator Takara Bio as per the manufacturer's protocol. Quantification of viral particles used p24 Rapid Titer Kit Takara Bio.

Ectopic expressions of CDH2 and Cx43 in MDA-MB-231 Oct4 GFP/CDH2-shRNA-RFP and MDA-MB-231 Oct4 GFP/Cx43-shRNA-RFP were done by spinfection. We transduced pLVX-TetOne-CDH2/GS and pLVX-TetOne-Cx43/GS at MOI of 2:1, with 4 µg/ml of polybrene. The cells were immediately centrifuged for 1 h at 1,200$g$ at 32°C for 1 h.

and presented at right, n = 3. **$P < 0.05$ versus Cx43 or CDH2 knockdown. **(E)** Cell viability was determined for MDA-MB-231, knockdown for CDH2 or Cx43, or control with scramble shRNA, treated with carboplatin (220 mg/ml). The analyses were performed at 24 h intervals up to 120 h using cell titer blue. The data are presented as % viable cells ± SD, n = 3. The percentage of viable cells at 120 h is zoomed on the right. *$P < 0.05$ versus knockdown cells at the 120 h time point. **(E, F)** The studies in "(E)" were repeated, except with doxorubicin (1 mM). *$P < 0.05$ versus Cx43 and CDH2 knockdown. **(G)** Protocol used to inject NSG mice, i.v. with 5 × 10$^5$ CSCs isolated from MDA-MB-231-Oct4-GFP, knockdown for CDH2 (RFP), Cx43 (RFP), or scrambled shRNA (RFP). Mice were injected intraperitoneally with 5 mg/ml carboplatin at days 3 and 5. **(H)** At day 7, sections from paraffin embedded femurs were labeled with anti–pan-cytokeratin-Texas Red (red) and anti–Ki67-AF405 (blue). Tissues were imaged with EVOS FL Auto 2 at magnification of 200×. Images represent six mice/group. **(I)** The number of BC cells (red) in mouse femur from "(H)" was counted using ImageJ and presented as mean BC cells ± SD, n = 6. **(J)** Cartoon summarizes the data in this Figure: Decreased apoptotic pathways in CSCs impart chemoreistance. Cx43/CDH2 knockdown cells reversed the resistance to chemosensitization. Source data are available for this figure.

After this, the cells were incubated at 37°C for 24 h. CDH2 expression was induced with 500 ng/ml Dox and Cx43 with 10 ng/ml Dox.

## Cell lines

MDA-MB-231 (Cat. no. HTB-26), T47D (Cat. no. HTB-133), HEK-293T (Cat. no. CRL-3216), and MCF12A (Cat. no. CRL-10782) were purchased from American Type Culture Collection (ATCC) and cultured as per their instruction. MDA-MB-231 is negative for estrogen, progester-one, and herceptin receptors (triple negative), whereas T47D is triple positive. MCF12A is a non-tumorigenic breast epithelial cell line.

We previously described MDA-MB-231 cells with stable trans-fection of pOct4a-GFP (Patel et al, 2012). Stably transfection of pEGFP1-Oct3/4 allowed for selection of CSC based on GFP intensity (top 5% of total cells). Hi, Med, and Low GFP represent the relative Oct4a expression. The pOct4a-GFP transfectants were stably knock-down for CDH2 and Cx43 by transfecting with pRFP-C-RS-Scramble, -CDH2, and -Cx43 using Lipofectamine 3000 as per the manufacturer's instruction. The stable transfectants were selected with puromycin. The media also contained G418 to ensure stability of pOct4a-GFP. The surviving colonies of cells were verified for CDH2 or Cx43 knockdown. Clone B was selected for CDH2 because of >80% knock-down efficiency (Fig S1F and G). We designated the stable knockdown cells as follows: M-Oct4-GFP/scramble-RFP, M-Oct4-GFP/CDH2-shRNA-RFP and M-Oct4-GFP/Cx43-shRNA-RFP.

CSC selection with CDH2 or Cx43 knockdown used the following gating schemes: We first selected the knockdown cells within the total BCCs by gating on RFP (knockdown and vector control), using forward (FSC-H), and side scatter (SSC-H). This was followed by a second gating for CSCs, based on GFP[hi]. We further ensure the selection of CSCs with knockdown Cx43 and CDH2 by a second gating of RFP[hi] cells.

## BM stromal culture

Stroma cells were cultured from BM aspirates, as described (Corcoran et al, 2007). The aspirate was diluted in $\alpha$-MEM containing 12.5% FCS, 12.5% horse serum, $10^{-7}$ mol/l hydrocortisone, $10^{-4}$ mol/l $\beta$-mercaptoethanol, and 1.6 mmol/l glutamine and then added to T25 tissue culture flasks. After 3 d, red blood cells and granulocytes were removed by Ficoll–Hypaque density gradient centrifugation. The mononuclear fraction was replaced in the flasks and at weekly intervals, 50% of media were replaced until confluence.

## MSC culture

MSCs were cultured from BM aspirates, as described (Corcoran et al, 2007). Briefly, BM aspirate was diluted in DMEM containing 10% FCS in vacuum gas plasma-treated 100 mm petri dishes (BD Falcon). The plates were incubated at 37°C with 5% $CO_2$. At day 3, red blood cells and granulocytes were removed by Ficoll–Hypaque density gra-dient centrifugation. The mononuclear fraction was replaced in the dishes. At weekly intervals, 50% of the media were replaced until the cells were ~80% confluent. At passage 3, the cells were char-acterized for phenotype and multipotency (Fig S1A–C).

## Flow cytometry

Cell surface labeling for CDH2 was performed with cells, washed with 1× PBS, and then fixed with 3.7% formaldehyde for at RT for 15 min. This was followed by labeling with anti–CDH2-PE for 30 min at RT, in the dark. In the case of intracellular labeling for CDH2, the fixed cells were washed with 1× PBS for 5 min and then per-meabilized at 4°C with 0.2% Triton X-100 for 5 min. After this, the cells were washed again with 1× PBS and then labeled with anti–CDH2-PE or APC as for extracellular labeling. The cells were washed with 1× PBS and then immediately analyzed on the FACSCalibur (BD Biosciences).

## Sorting of BCC subsets

BCC subsets were sorted as described (Patel et al, 2012). Briefly, BCCs, stably transfected with pEGFP1-Oct3/4, were selected, based on relative GFP expression using the FACSDiva (BD Biosciences). The top 5% GFP (Oct4[hi]) cells contained CSCs (Patel et al, 2012). Further sorting was conducted for CSCs with specific knockdown, based on the fluorescence-tagged shRNA, or scramble vector: GFP[hi]/RFP[hi]: MDA-MB-231 Oct4 GFP/scramble RFP, MDA-MB-231 Oct4 GFP/CDH2-shRNA-RFP, and MDA-MB-231 Oct4 GFP/Cx43-shRNA-RFP cells were sorted on the FACSDiva (BD Biosciences) to select GFP[hi]/RFP[hi] cells.

## Dye transfer assay

GJIC between CSCs and stroma or MSCs were performed as de-scribed by Patel et al (2012). MSCs or stromal cells were labeled with 2.5 mM of Cell Tracker Blue CMAC Dye (microscopy) or Cell Tracker Deep Red dye (flow cytometry). For labeling the cells, the dyes were incubated for 30 min at 37°C in a $CO_2$ incubator. After incubation, the cells were washed twice with 1× PBS. Co-cultures of dye-loaded MSCs or stromal fibroblasts and CSCs (yellow: Oct4A-GFP[hi] + shRNA/RFP[hi]) at 1:1 ratio were assessed for GJIC by imaging for dye transfer into CSCs. Control co-cultures used CSCs with scramble shRNA BCCs with vehicle or 1-octanol (300 μM). Dye transfer by flow cytometry occurred after 72 h as follows: Cells were trypsinized, pelleted by centrifugation, and then resuspended in 0.5 ml PBS. Fluorescence microscopy: Cells were imaged using the EVOS FL Auto 2. The number of CSCs with dye were counted 10 fields per condition using ImageJ (Schneider et al, 2012) (n = 3) (https://imagej.net/ RRID: SCR_003070). Percent of dye transfer is number of CSCs with dye divided by total number of cells within the field, times 100.

## Immunocytochemistry

GFP[hi]/RFP[hi] sorted MDA-MB-231 were placed in four-well chamber slides (Nunc; Thermo Fisher Scientific). After 24 h, the cells adhered and were fixed with 3.7% formaldehyde for 15 min at RT followed by permeabilization for 5 min using 0.2% Triton X-100 in PBS. The cells were blocked with 1% BSA in PBS for 1 h. The wells were carefully washed twice with 1X PBS and then incubated for 2 h at RT with the following: anti-Cx43 (1/200), anti-CDH2-PE (1/20), anti-GM130 (1/200), or anti-GRP78 (1/200). All antibodies were diluted in 1× PBS. After incubation, the wells were washed with 1× PBS and then incubated with goat anti-mouse IgG–Alexa Fluor 488 and goat anti-

rabbit IgG–Alexa Fluor 405. The cells were imaged with the EVOS FL Auto 2.

### Golgi disruption assay

MDA-MB-231 were plated at $2 \times 10^4$ in four-well chamber slides. After overnight adherence, the cells were treated with 5 µg/ml of brefeldin-A. After 24 h, the cells were washed and then fixed, followed by labeling with anti-CDH2 and anti-Cx43 as above.

## Image stream multispectral imaging

Image stream multispectral analyses used ImageStreamX Mark II Imaging Flow Cytometer, as described (Ghazaryan et al, 2014). Briefly, MDA-MB-231 cells were trypsinized, pelleted, resuspended for fixing, and permeabilization with 3.7% formaldehyde for 15 min and 0.2% Triton X-100 in PBS for 5 min, respectively. The cells were incubated with anti–CDH2-PE (1:20) for 30 min in dark followed by washing with 1× PBS. After this, the cells were resuspended in 500 µl of 1% formaldehyde in 1× PBS. Cx43 was assessed by labeling with goat anti-rabbit Cx43 (1:100) for 30 min at RT and secondary goat anti-rabbit Alexa 610 IgG for another 30 min in the dark. The cells were then washed with 1× PBS and resuspended in 500 µl of 1% formaldehyde in 1× PBS. The cells were acquired with the Image-StreamX Mark II Imaging Flow Cytometer (Luminex).

## Single cell imaging

Single cell imaging for colocalized proteins used the Nanoimager S Mark II from ONI (Oxford Nanoimaging) with lasers 405 nm/150 m, 488 nm/1 W, 561 nm/500 mW, and 640 nm/1 W. Dual channel acquisition was setup using 640-nm dichoric mirror paired with 576–620-nm bandpass filter for PE and 685/40 nm emission filters for Alexa 647. Images were acquired with NimOS 1.3.7511 imaging software at 1,000× magnification and 1.4 NA objective. Four color wide-field images were acquired in a sequential mode with 100-ms exposure time for each image. To show protein localization, all the channels were reconstructed in Fiji.

## Immunoprecipitation

Cells were lysed as for Western blot, and 15-µg protein extracts used to immunoprecipitate CDH2 and Cx43, as described (Ghazaryan et al, 2014). Briefly, the lysates were precleaned with magnetic beads bound to IgG. During this time, the extracts were incubated on ice for 30 min. After this, the beads were magnetically removed and the extracts were incubated overnight at 4°C with 2 µg/ml goat anti-rabbit Cx43 from Cell Signaling Technology, 2 µg/ml goat anti-rabbit CDH2, or 0.5 µg/ml rabbit anti-human Flag tag or normal anti-rabbit IgG. After 24 h, the extracts were incubated with magnetic beads bound to anti-rabbit IgG for 4 h/4°C. The unbound protein was then removed using a magnetic separator. The bead bound proteins were then washed five times using lysis buffer and then resuspended in 2× loading dye with reducing agent, β-mercaptoethanol. The immunoprecipitated lysates were electrophoresed on 12% SDS–PAGE and then developed for CDH2, Cx43, Flag, or HA with specific antibodies.

## Molecular modeling

The molecular interaction between CDH2-Cx43 was predicted using ZDOCK (http://zdock.umassmed.edu/) and functional interaction using STRING (v11) (https://string-db.org/ RRID:SCR_005223) (Pierce et al, 2014; Szklarczyk et al, 2019). ZDOCK used CDH2 (PDB ID: 3Q2W, ectodomain) and Cx43 (PDB ID: 2LL2, cytoplasmic domain) X-ray crystallographic database ID number as the protein of interest for input. CDH2 protein sequence is evolutionarily conserved, hence, identical sequences in mouse and human. As such, the murine crystal structure was used for ZDOCK analyses. The top five potential binding sites are shown in grey for CDH2 and blue, green, orange, pink, and red for Cx43 (Fig 3C). STRING used Cx43 (GJA1) as input, resulting in CDH2 as one of the potential interacting proteins.

## Monocyte/MΦ culture

M2 MΦ were isolated from monocytes as described (Walker et al, 2019). Briefly, the monocytes were selected from human PBMCs using human CD14+ Dynabeads Flow Kit (Thermo Fisher Scientific). The monocytes were incubated with 50 ng/ml M-CSF. After 48 h, we replaced the media with fresh medium containing 20 ng/ml IL-4. After 3 d, the cells were positive for CD206 and negative for MHC-II.

## In vivo studies

The Institutional Animal Care and Use Committee of Rutgers, Newark Campus, approved the use of mice. Female (6 wk) NOD/SCID (NSG) mice were purchased from Jackson Laboratories and housed at an AAALAC-accredited facility, Rutgers Comparative Medicine Resource, New Jersey Medical School, Newark, NJ.

### BC dormancy

We established dormancy in female nude and NOD/SCID (NSG) BALB/c mice, as described (Patel et al, 2012; Bliss et al, 2016; Walker et al, 2019). Briefly, we sorted CSCs from BCCs with stable pOct4a-GFP with or without Cx43/CDH2-shRNA. Control cells expressed pOct4a and scramble shRNA. We previously reported on self-renewal of the Oct4a[hi] BCCs using in vitro and in vivo studies, including serial passaging (Patel et al, 2012). We injected $5 \times 10^5$ Oct4a[hi] sorted BCCs with the highest GFP (Oct4a[hi]) intravenously via the tail vein of mice. After sorting to injecting of the CSCs, it is possible that there could be differentiation of the CSCs, resulting in proliferating cells. Thus, to ensure elimination of these cells, we injected the mice at days 3 and 5 with low-dose carboplatin (2 mg/kg), intraperitoneally. The additional chemotherapy did not change the CDH2, Cx43, or Oct4a expression on the CSCs. These genes were the highest in CSCs, relative to the other BC subsets (Fig 1). Dormancy in mice femurs was sustained even without the low dose chemotherapy, which was used to eliminate proliferating cells.

## CDH2 and Cx43 knockdown BCCs

NSG mice were injected with $5 \times 10^5$ Oct4a[hi] MDA-MB-231 Oct4 GFP/scramble RFP, MDA-MB-231 Oct4 GFP/CDH2-shRNA-RFP, and MDA-MB-231 Oct4 GFP/Cx43-shRNA-RFP CSCs intravenously (tail vein). At day 10, mice were euthanized and embedded in parafilm for

processing, as described above. The experiments with NSG mice were repeated by co-injecting $10^6$ CMAC (blue)-labeled M2Φ and the following CSCs isolated from MDA-MB-231-Oct4-GFP: scramble-RFP, CDH2-shRNA-RFP, or Cx43-shRNA-RFP. At day 10, the mice were euthanized and tissues processed as for nude mice.

## Immunohistochemistry

Antigens were retrieved from parafilm-embedded mouse femurs and human BM biopsies by overnight incubation at 56°C. After this, the slides were dewaxed with xylene and ethanol and then dehydrated, followed by rehydration using deionized water. The cells were permeabilized with 0.1% Triton X-100. The slides were then washed with 1X PBS and then incubated overnight at 37°C in a humidified chamber with primary antibodies, each at 1/250 final dilution. The slides were washed thrice with 1× PBS then incubated for 2 h at RT in a humidified chamber in the dark with fluorescence-tagged secondary antibodies (1/500 final dilution). After washing the slides with 1× PBS, the tissue was immediately analyzed on the EVOS FL Auto 2 Imaging System. The results of immunohistochemical analyses of slides from mice and human sections were quantified with ImageJ. We counted BCCs in 10 fields/slide. Immunohistochemistry for human pan-cytokeratin and Ki67 was performed in murine femurs and CDH2, Cx43, and pan-cytokeratin for human BM biopsies.

## PLA

Interaction between CDH2 and Cx43 was confirmed in human BM BC biopsy (randomly selected patient 3) using PLA as per manufacturers instruction (Alam, 2018). Briefly, the antigen surface of parafilm-embedded biopsy was exposed, as explained before in immunohistochemistry. Anti-CDH2 and anti-Cx43 PLA probe was first created followed by ligation and signal amplification using Texas red probe. The signal was amplified only when the two proteins are <40 nm apart. The slides were imaged using EVOS FL Auto 2 at 600× magnification. Isotype was negative control probe created as per manufacturer's instruction.

## RNA sequencing (Seq)

CSCs from MDA-MB-231 Oct4 GFP/scramble RFP, MDA-MB-231 Oct4 GFP/CDH2-shRNA-RFP, and MDA-MB-231 Oct4 GFP/Cx43-\shRNA-RFP were sorted as outlined in Fig 1E (Patel et al, 2012). The cells were allowed to recover from stress by incubating overnight at 37°C, 5% $CO_2$. Total RNA was isolated with RNeasy Mini Kit (QIAGEN) and then submitted to the Genomics Center at Rutgers New Jersey Medical School. Ribosomal RNA was depleted with Ribo-Zero gold from Illumina and then used to prepare libraries for next generation sequencing. The libraries were prepared with NEB Ultra II Library Preparation kit and NEBNext Multiplex Oligos for Illumina (Dual Index Primers Set 1) (New England BioLabs). Upon quality control of the libraries with Qubit instrument and a high-sensitivity Kit from Thermo Fisher Scientific as well as TapeStation 2200 instrument and D1000 ScreenTapes from Agilent, the cDNA libraries were diluted to 2 nM. The libraries were denatured as per Illumina's protocol and then sequenced on Illumina's NextSeq 500

[NIH-1S10OD018206-01A1] using 1X75 cycle high output kit. The BCL files from sequencing were converted to FastQ files using BCL2FASTQ software from Illumina (RRID:SCR_015058).

## Data analyses

### RNA-Seq

Partek Flow software (Version 7) was used to convert fastq.gz to txt files (Partek Inc.). Reads were then aligned using STAR 3.6.1 to hg38 with Partek's optimization of the expectation–maximization algorithm. One low-quality sample containing more than 50% intergenic and intronic reads was removed and the remainder of the data were normalized using log counts per million +1. Batch effect among three replicates of sequencing was removed via Partek log normal model with shrinkage and low count reads were removed (maximum sample <1.5).

Partek GSA analysis was used to interrogate the effect of the two knockdowns versus scramble (control). Genes with a false discovery rate < 0.05 and a fold change of 1.66 in shRNA CDH2 versus Scramble and a fold change of 1.5 in shRNA Cx43 versus Scramble were retained. Differentially expressed genes were visualized in a heat map with hierarchical clustering in Morpheus (Broad Institute: https://software.broadinstitute.org/morpheus). Common dendrogram nodes were queried for enriched gene ontology pathways and subsequently annotated.

GSEA analyses were also used on all genes that passed quality control. Hallmark and C2 gene sets were used for analysis. Significant results, q < 0.05, were retained. Ingenuity Pathway Analysis (IPA; QIAGEN Inc.) of differentially expressed genes was used to interrogate the effect on canonical pathways and cell functions, as well as to generate networks of selected cell functions. The pathways passing $P < 0.01$ were ranked using $-\log_{10} P$-value times activation z-score.

## Real-time PCR

Real-time PCR for CDH2 was performed as described (Patel et al, 2012). Briefly, RNA was extracted from samples using TRIzol as per the manufacturer's instruction. RNA was quantified with QIAxpert and then converted to cDNA with the High-Capacity cDNA Reverse Transcription Kit. Real-time PCR used Platinum SYBR Green qPCR SuperMix-UDG Kit and the 7300 Real-Time PCR System (Thermo Fisher Scientific). The following primers were used in the PCR mix: CDH2 forward, 5′-CAG TGG CCA CCT ACA AAG-3′; CDH2 reverse, 5′-AAA TGA AAC CGG GCT ATC-3′; β-actin forward, 5′-GCC CTA TAA AAC CCA GCG GC-3′; β-actin reverse, 5′-AGA GGC GTA CAG GGA TAG CA-3′; Human GAPDH forward, 5′-CAG AAG ACT GTG GAT GGC C-3′; Human GAPDH reverse, 5′-CCA CCT TCT TGA TGT CAT C-3′; Human and mouse GAPDH, forward 5′-AGT CCC CCA CCA CAC CTG AT-3′ and human and mouse GAPDH reverse, 5′-TTG ATG GTA CAT GAC AAG GTG C-3′.

## Western blot analyses

The proteins were isolated using a cell lysis buffer (50 mM Tris–HCl [pH 7.4], 100 mM NaCl, 2 mM $MgCl_2$, 10% glycerol, and 1% NP-40) as explained in Ghazaryan et al (2014). The extracts (15 μg) were electrophoresed on 12% SDS–PAGE gel. Proteins were transferred

onto polyvinylidene difluoride (PVDF) membranes (Perkin Elmer). The membranes were incubated overnight at 4°C on rocker with anti-Cx43, anti-CDH2, anti-Vimentin, anti-Twist, anti-Slug, anti-$\beta$–catenin, anti-Zeb1, anti-caspase 3, anti-caspase-7, anti-MDR1, or anti-$\beta$-actin at a dilution mentioned above in 1% nonfat milk dissolved in 1× PBS Tween. Next day, the membranes were incubated with species-specific HRP-tagged IgG in 1% milk 1× PBS Tween for 2 h at 4°C. The membranes were developed chemiluminescence using SuperSignal West Femto Maximum Sensitivity Substrate. The band densities were normalized using UN-SCAN-IT densitometry software (RRID:SCR_017291; Silk Scientific).

### Tumorsphere assay and in vitro serial passage

Tumorsphere assay used CSCs from MDA-MB-231-Oct4-GFP with scramble-RFP, CDH2-shRNA-RFP or Cx43-shRNA-RFP, as described (Patel et al, 2012). Cells were seeded 1 cell/well in 96-well low adhesion plate (Costar). After 10 d, tumorspheres were dissociated with trypsin and then passed through a 40 mm mesh (BD cell strainer cap tube) and the tumorsphere assay repeated five times (serial passaging).

### Wound healing (scratch) assay

MDA-MB-231-pOct4A-GFP, knockdown for Cx43 or CDH2 or transfected with scrambled shRNA, were seeded at $5 \times 10^5$ cells/well in six-well plates. The plates were incubated overnight at 37°C and 5% $CO_2$. A scratch was formed across the confluent cells with a sterile 200 µl tip. After this, the media were replaced with fresh media. The scratch area was imaged using the EVOS Auto FL2 at times ranging between 1 and 50 h.

The wound size was quantified by measuring the distance between the scratch. Time 0 wound was considered to be 100% gap and was given the value of 1. Wound was calculated by measuring the gap at a given time point/the original size of gap at time 0. The distance between the scratch was measured using ImageJ software. The experiment was repeated three times (n = 3). The data were analyzed using $t$ test; $P$-value ≤ 0.05 was considered to be significant.

### Cell cycle analyses

Cell cycle analyses colabeled cells with 7AAD and PyY, as described (Bliss et al, 2016). Because PyY and RFP from shRNA used same channel on the flow cytometry, we transiently transfected the cells with siRNA using MDA-MB-231 and T47D. The cells were transfected with CDH2 siRNA or Cx43 siRNA or Risc free (control) siRNA using Lipofectamine RNAimax. After 24 h, the cells were labeled with PyY followed by 7AAD staining. The cells were labeled with 7-AAD and PyY and gated for low DNA based on 7-AAD and then selected the labeled RNA, based on PyY incorporation for assessing the $G_0/G_1$ population.

### Cell viability

Cell viability was performed as described (Patel et al, 2012). M-Oct4-GFP/scramble-RFP, M-Oct4-GFP/CDH2-shRNA-RFP, and M-Oct4 GFP/Cx43-shRNA-RFP cells were treated with carboplatin (220 µg/ml) or doxorubicin (1 µM). The cell viability was assessed every 24 h with CellTiter-Blue as per the manufacturer's instruction. The fluorescence was taken at $560(20)_{Ex}/590(10)_{Em}$.

The absorbance of each well at 570 and 600 nm wavelength was used to calculate viability as follows:

$$\frac{\text{Treated } (\text{Absorbance}_{570}-\text{Absorbance}_{600}) \times 100}{\text{Average of Untreated } (\text{Absorbance}_{570}-\text{Absorbance}_{600})}$$

The data were analyzed using $t$ test; $P$-value ≤ 0.05 was considered to be significant.

### Reporter gene assay

Reporter gene assays were performed as described (Qian et al, 2001). M-Oct4 GFP cells with CDH2 or Cx43 knockdown or with scrambled shRNA were transfected with human cyclin D1 promoter in pGL3-basic. The cells were transfected with Lipofectamine 3000, as per the manufacturer's instruction. The cells were co-transfected with p$\beta$gal as described (Qian et al, 2001). After 24 h, cell lysates were prepared using 1× Promega lysis buffer. Protein concentration was determined using the Bradford protein assay reagent. Luciferase and $\beta$-galactosidase levels were quantified using kits from Promega. For calculating relative luminescence unit, the luciferase values were normalized with $\beta$-galactosidase.

### Apoptosis assay

We quantitated apoptotic activity with Apo-ONE Homogeneous Caspase 3/7 assay kit. The method followed the manufacturer's instruction. Briefly, M-Oct4-GFP, knockdown for CDH2 or Cx43 or scramble shRNA were added to 96-well black plates. Each experimental point used six replicates. Apo-ONE caspase 3/7 reagent was added to each well. The plate was incubated for 4 h at RT, followed by fluorescence measurement at $499_{Ex}/521_{Em}$. The relative fluorescence unit was calculated by subtracting relative fluorescence unit blank from the samples. The same assay was repeated with carboplatin (220 µg/ml) and doxorubicin (1 µM) treated BCCs.

## Data Availability

The data from Next Generation Sequencing are available in the Gene Expression Omnibus database: GSE138434 (https://www.ncbi.nlm.nih.gov/geo/query/acc.cgi?acc=GSE138434).

## Supplementary Information

## Acknowledgements

This work was supported in part by a grant awarded by the New Jersey Commission of Health and Metavivor Foundation. This work is in partial fulfillment for a doctoral thesis for G Sinha.

## Author Contributions

G Sinha: conceptualization, data curation, software, validation, methodology, and writing—original draft.
AI Ferrer: software, methodology, and writing—review and editing.
S Ayer: software, formal analysis, methodology, and writing—review and editing.
MH El-Far: software, validation, methodology, and writing—review and editing.
SH Pamarthi: validation, methodology, and writing—review and editing.
Y Naaldijk: data curation, software, methodology, and writing—review and editing.
P Barak: resources, software, investigation, visualization, and writing—review and editing.
OA Sandiford: resources, validation, methodology, and writing—review and editing.
BM Bibber: conceptualization, data curation, methodology, and writing—review and editing.
G Yehia: validation, methodology, and writing—review and editing.
SJ Greco: resources, formal analysis, validation, methodology, and writing—review and editing.
J-G Jiang: data curation, formal analysis, visualization, and writing—review and editing.
M Bryan: resources, data curation, investigation, and writing—review and editing.
R Kumar: data curation, formal analysis, methodology, and writing—review and editing.
NM Ponzio: formal analysis, supervision, methodology, and writing—review and editing.
J-P Etchegaray: validation, methodology, and writing—review and editing.
P Rameshwar: conceptualization, data curation, formal analysis, supervision, funding acquisition, project administration, and writing—review and editing.

## Conflict of Interest Statement

The authors declare that they have no conflict of interest.

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
