## [Reviewer comments · Life Science Alliance]

Life Science Alliance

Specific N-cadherin-dependent pathways drive human breast cancer dormancy in bone marrow

Garima Sinha, Alejandra I. Ferrer, Seda Ayer, Markos El-Far, Sri Harika Parmarhi, Yahaira Naaldijk, Pradeep Barak, Oleta Sandiford, Bernadette Bibber, Ghassan Yehia, Steven Greco, Jie-Gen Jiang, Margarete Bryan, Rakesh Kumar, Nicholas Ponzio, Jean-Pierre Etchegaray, and Pranela Rameshwar

DOI: <https://doi.org/10.26508/lsa.202000969>

Corresponding author(s): Pranela Rameshwar, New Jersey Medical School, Rutgers

Review Timeline:

Submission Date:	2020-11-20
Editorial Decision:	2021-01-06
Revision Received:	2021-03-30
Editorial Decision:	2021-04-13
Revision Received:	2021-04-19
Accepted:	2021-05-20

Scientific Editor: Shachi Bhatt

Transaction Report:

January 6, 2021

Re: Life Science Alliance manuscript #LSA-2020-00969-T

Dr. Pranela Rameshwar
New Jersey Medical School, Rutgers
Medicine
185 South Orange Avenue
MSB, Room E-579
Newark, NJ 7103

Dear Dr. Rameshwar,

Thank you for submitting your manuscript entitled "Specific N-cadherin-dependent pathways drive human breast cancer dormancy in bone marrow" to Life Science Alliance. The manuscript was assessed by expert reviewers, whose comments are appended to this letter.

As you can see from the reviewers' comments, the reviewers are quite enthusiastic about these findings, but have also pointed out a number of reasonable revisions to improve the solidity of the study. We would thus like to invite you to submit a revised version on the manuscript that addresses all of the reviewers' concerns.

Thank you for this interesting contribution to Life Science Alliance. We are looking forward to receiving your revised manuscript.

Sincerely,

Shachi Bhatt, Ph.D.
Executive Editor
Life Science Alliance
<https://www.lsjournal.org/>
Tweet @SciBhatt @LSAJournal

- A letter addressing the reviewers' comments point by point.
- An editable version of the final text (.DOC or .DOCX) is needed for copyediting (no PDFs).
- High-resolution figure, supplementary figure and video files uploaded as individual files: See our detailed guidelines for preparing your production-ready images, <https://www.life-science-alliance.org/authors>
- Summary blurb (enter in submission system): A short text summarizing in a single sentence the study (max. 200 characters including spaces). This text is used in conjunction with the titles of papers, hence should be informative and complementary to the title and running title. It should describe the context and significance of the findings for a general readership; it should be written in the present tense and refer to the work in the third person. Author names should not be mentioned.

B. MANUSCRIPT ORGANIZATION AND FORMATTING:

Reviewer #1 (Comments to the Authors (Required)):

Sinha et al., characterize the mechanisms of breast cancer cell dormancy by focusing on two proteins that are associated with establishing a link between cancer cells and bone marrow niche through gap junction. Authors focused on the molecule called N-Cadherin and on connexin 43. Authors conclude that these proteins are associated with conferring cancer stem cell phenotype to dormant cells and restrict their proliferation. Loss of their expression leads to switch in organotropism of metastatic cancer cells as cancer cells depleted of N-cadherin and connexin 43 now metastasize to lungs and brain.

Overall the study is very comprehensive and reveals novel function of N-cadherin independent of its known role in EMT. Although the majority of in vivo studies was done using a single cell line, MDA-MB-231, it is unfortunate limitation of this field, as not many breast cancer cell lines reliably metastasize to bone. The manuscript can be further strengthened if authors can address the following comments.

- 1) Does knockdown of CDH2 or CX43 leads to upregulation of E-cadherin, particularly in CSCs, which makes cancer cells more susceptible to lung and brain metastasis?
- 2) Authors need to measure ZEB1 levels in CSCs upon knockdown of CDH2 and CX43 as ZEB1 plays critical role in CSC phenotype of certain cell types (Figure 5H).
- 3) P38 kinase pathway is one of the extensively studied signaling network in tumor dormancy, although not specific to bone. Is there an effect of CDH2 and CX43 knockdown in p38 kinase pathway?
- 4) Could therapeutic targeting of this axis lead to increased metastasis to lungs and brain compared to keeping cells dormant in bone? Although knockdown cells are more sensitive to certain chemotherapy, metastatic niche in brain and lungs may protect these cells from the drugs. Authors need to discuss this limitation.
- 5) In many places, figure numbers are in correct in the results section. Figure 3C in page 11 should 3B and 3D should 3C. In page 15, S7G/H should be S5G/H.
- 6) Proofreading of the manuscript is required. Few examples are: In summary, we knockdown of CDH2 in CSCs and demonstrated reduced cells in femurs- page 21. We asked of co-localized CDH2 and CX43 directly interact (page 11).
- 7) Discussion can be shortened as significant fraction contains summary of results

Reviewer #2 (Comments to the Authors (Required)):

This manuscript shows data that dormant cancer stem cells of the breast metastasize to the bone marrow and form Cx43-mediated gap junctions. The authors describe the significance of CDH2 in Cx43-dependent dormancy of breast cancer cells in the bone marrow. The authors postulate methods by which this interaction can be disrupted such that breast cancer cells can come out of dormancy and become susceptible to chemo.

Overall, the conclusions made by the authors are strongly supported by the data presented. The scientific experiments are quite rigorous. The data is robust for all assays and methodology is sound. Specifically, the combination of novel laboratory data and bioinformatics supports the conclusions.

- 1) Please explain the role of 1-octanol in the study.
- 2) On page 10, Please explain further about the rationale for looking at co-localization of Cx43 and CDH2 and why the plan was to look at specific organelles.
- 3) For the apoptosis assay, please explain why carboplatin was used, since breast cancer is mostly treated with adriamycin and cytoxan.
- 4) please elaborate further on the translational relevance with regards to inhibition of Cx43 or CDH2.

March 29, 2021

Shachi Bhatt, Ph.D.
Executive Editor
Life Science Alliance

RE: Life Science Alliance manuscript #LSA-2020-00969-T

Dear Dr. Bhatt,

We thank you, the journal staff and reviewers for the time spent on the referenced manuscript. We have addressed each comment, point by point, **shown in red text**.

Reviewer #1:

Comment 1: Does knockdown of CDH2 or CX43 leads to upregulation of E-cadherin, particularly in CSCs, which makes cancer cells more susceptible to lung and brain metastasis?

Response 1: We performed new data with Western blot using CSC extracts. The results showed undetectable E-cadherin. The control studies with MCF12A extracts were positive. The data shown in Fig. 5H is presented in the text, last sentence of the first paragraph, p. 14. We cannot conclude that E-cadherin was not involved. Going forward, we will test this with less aggressive cell lines and even primary cells.

Comment 2: Authors need to measure ZEB1 levels in CSCs upon knockdown of CDH2 and CX43 as ZEB1 plays critical role in CSC phenotype of certain cell types (Figure 5H).

Response 2: We analyzed CSC extracts for Zeb1 by Western blot. The CSCs were transfected with scramble shRNA or, knockdown for Cx43 or CDH2. The results are included in Fig. 5G with the corresponding description on p. 14, end of Paragraph 1.

Comment 3: P38 kinase pathway is one of the extensively studied signaling network in tumor dormancy, although not specific to bone. Is there an effect of CDH2 and CX43 knockdown in p38 kinase pathway?

Response 3: Thanks for this comment – Indeed, knockdown of both CDH2 and Cx43 led to increased cell proliferation and expression/repression of the associated genes (Figs 5C/D). The newly inserted Western blot for total and phosphorylated Erk 1/2 and p38 showed decrease in Erk1/2 with the extracts from knockdown cells (Fig. 5I). We revised the results section accordingly as well as the conclusion, end p14-p.15.

Comment 4: Could therapeutic targeting of this axis lead to increased metastasis to lungs and brain compared to keeping cells dormant in bone? Although knockdown cells are more sensitive to certain chemotherapy, metastatic niche in brain and lungs may protect these cells from the drugs. Authors need to discuss this limitation.

Response 4: We discussed these findings in the context of how the data could be expanded towards treatment (p. 18).

Comment 5: In many places, figure numbers are in correct in the results section. Figure 3C in page 11 should 3B and 3D should 3C. In page 15, S7G/H should be S5G/H.

Response 5: We have carefully reviewed the text to accurately cite the appropriate Figures.

Comment 6: Proofreading of the manuscript is required. Few examples are: In summary, we knockdown of CDH2 in CSCs and demonstrated reduced cells in femurs- page 21. We asked of co-localized CDH2 and CX43 directly interact (page 11).

Response 6: The authors have carefully reviewed the text to correct grammatical errors, incomplete sentences and other typographical errors.

Comment 7: Discussion can be shortened as significant fraction contains summary of results.

Response 7: We avoided repeating the results in the discussion section unless specific findings were required in the discussion.

Reviewer #2:

Comment 1: Please explain the role of 1-octanol in the study.

Response 1: We revised the results section to clarify the use of 1-Octanol in the experiments, top of p. 9.

Comment 2: On page 10, Please explain further about the rationale for looking at co-localization of Cx43 and CDH2 and why the plan was to look at specific organelles.

Response 2: We revised the entire results section relative to intracellular co-localization of Cx43 and CDH2 (Figs. 2C-2E). The response to specific comment 2 is included in red text, p. 10.

Comment 3: For the apoptosis assay, please explain why carboplatin was used, since breast cancer is mostly treated with adriamycin and cytoxan.

Response 3: The studies were performed with triple negative and positive CSCs, knockdown for Cx43 or CDH2. Carboplatin allowed us to study both cell types with the same chemotherapy. This information is included on the end of p. 16.

Comment 4: please elaborate further on the translational relevance with regards to inhibition of Cx43 or CDH2.

Response 4: We elaborated on the translational relevance of the findings, p. 18.

We thank you for taking giving us the opportunity to revise our manuscript.

Sincerely,

Pranela Rameshwar

Pranela Rameshwar, Ph.D.
Professor, Rutgers, New Jersey Medical School
Tel. (973) 972 0625
Email: rameshwa@njms.rutgers.edu

April 13, 2021

RE: Life Science Alliance Manuscript #LSA-2020-00969-TR

Dr. Pranela Rameshwar
New Jersey Medical School, Rutgers
Medicine
185 South Orange Avenue
MSB, Room E-579
Newark, NJ 7103

Dear Dr. Rameshwar,

Thank you for submitting your revised manuscript entitled "Specific N-cadherin-dependent pathways drive human breast cancer dormancy in bone marrow". We would be happy to publish your paper in Life Science Alliance pending final revisions necessary to meet our formatting guidelines.

Along with the points listed below, please also attend to the following:

- please add a Category for your manuscript in our system
- please use the [10 author names, et al.] format in your references (i.e. limit the author names to the first 10)
- please add your table legends to the main manuscript text after the references section, as well
- there are callouts for Figure S7G, H; S3D - Please correct
- please add callouts for Figures S3B; S5G, H, and I to your main manuscript text
- please upload your Tables in editable .doc or excel format
- please upload both main and supplementary figures as single files
- we can see that you have included the RNASeq accession number in the 'Data Analysis' section. We would request you to move it to a separate 'Data Availability section' for easier access to the readers
- please improve the size and visibility of the scale bar numbers in Figures 2E, 3A, I, S2, S3A, B, and scale bars in Figure 5L
- Inset panels in Figures 1G, H, and S1J, I are the same. It is LSA's policy to not allow for duplicated figures in the manuscript. We suggest you replace the panels in one of those figures with new ones
- Please provide an experimental duplicate for the WBs in Figure 5G and for Caspase 7/beta-actin WB in Figure 7B

To avoid unnecessary delays in the acceptance and publication of your paper, please read the

following information carefully.

A. FINAL FILES:

B. MANUSCRIPT ORGANIZATION AND FORMATTING:

Sincerely,

Shachi Bhatt, Ph.D.
Executive Editor
Life Science Alliance
<http://www.lsajournal.org>
Tweet @SciBhatt @LSAJournal

Reviewer #1 (Comments to the Authors (Required)):

This is the revised manuscript, which I reviewed before. Authors have addressed concerns raised in the last review through additional experiments. Results are compatible with overall conclusions

May 20, 2021

RE: Life Science Alliance Manuscript #LSA-2020-00969-TRR

Dr. Pranela Rameshwar
New Jersey Medical School, Rutgers
Medicine
185 South Orange Avenue
MSB, Room E-579
Newark, NJ 7103

Dear Dr. Rameshwar,

Thank you for submitting your Research Article entitled "Specific N-cadherin-dependent pathways drive human breast cancer dormancy in bone marrow". It is a pleasure to let you know that your manuscript is now accepted for publication in Life Science Alliance. Congratulations on this interesting work.

DISTRIBUTION OF MATERIALS:

Again, congratulations on a very nice paper. I hope you found the review process to be constructive and are pleased with how the manuscript was handled editorially. We look forward to future exciting submissions from your lab.

Sincerely,

Shachi Bhatt, Ph.D.

Executive Editor

Life Science Alliance

<http://www.lsajournal.org>
